# Breunnerite grain and magnesium isotope chemistry reveal cation partitioning during aqueous alteration of asteroid Ryugu

Toshihiro Yoshimura [1,16] ✉, Daisuke Araoka [2], Hiroshi Naraoka[3], Saburo Sakai[1], Nanako O. Ogawa [1], Hisayoshi Yurimoto [4], Mayu Morita [5], Morihiko Onose[5], Tetsuya Yokoyama [6], Martin Bizzarro[7], Satoru Tanaka[5], Naohiko Ohkouchi[1], Toshiki Koga [1], Jason P. Dworkin [8], Tomoki Nakamura[9], Takaaki Noguchi [10], Ryuji Okazaki[3], Hikaru Yabuta [11], Kanako Sakamoto [12], Toru Yada[12], Masahiro Nishimura[12], Aiko Nakato [12], Akiko Miyazaki [12], Kasumi Yogata [12], Masanao Abe[12], Tatsuaki Okada [12], Tomohiro Usui[12], Makoto Yoshikawa [12], Takanao Saiki[12], Satoshi Tanaka[12], Fuyuto Terui[13], Satoru Nakazawa [12], Sei-ichiro Watanabe [14], Yuichi Tsuda[12], Shogo Tachibana [12,15] & Yoshinori Takano [1,16] ✉

Returned samples from the carbonaceous asteroid (162173) Ryugu provide pristine information on the original aqueous alteration history of the Solar System. Secondary precipitates, such as carbonates and phyllosilicates, reveal elemental partitioning of the major component ions linked to the primordial brine composition of the asteroid. Here, we report on the elemental partitioning and Mg isotopic composition ($^{25}Mg/^{24}Mg$) of breunnerite [(Mg, Fe, Mn) $CO_3$] from the Ryugu C0002 sample and the A0106 and C0107 aggregates by sequential leaching extraction of salts, exchangeable ions, carbonates, and silicates. Breunnerite was the sample most enriched in light Mg isotopes, and the $^{25}Mg/^{24}Mg$ value of the fluid had shifted lower by ~0.38‰ than the initial value (set to 0‰) before dolomite precipitation. As a simple model, the $Mg^{2+}$ first precipitated in phyllosilicates, followed by dolomite precipitation, at which time ~76−87% of $Mg^{2+}$ had been removed from the primordial brine. A minor amount of phyllosilicate precipitation continued after dolomite precipitation. The element composition profiles of the latest solution that interacted with the cation exchange pool of Ryugu were predominantly Na-rich. $Na^+$ acts as a bulk electrolyte and contributes to the stabilization of the negative surface charge of phyllosilicates and organic matter on Ryugu.

Samples from the carbonaceous asteroid (162173) Ryugu are the freshest among extraterrestrial carbonaceous materials[1–3]. Ryugu's chemical composition, which corresponds to an Ivuna-type carbonaceous (CI) chondrite, suggests that a unique combination of secondary minerals formed by aqueous alteration of the parent body[4–7].

Aqueous alteration of carbonaceous chondrites is caused by the melting of ice accreted to the host rock by heating from radioactive decay of radioisotopes such as $^{26}Al$, and in many cases, multi-stage formation events of phyllosilicates and carbonates occur over multiple generations as a result of water–rock reactions[8,9]. The mineral

composition, isotopic ratios, and fluid inclusions of these altered minerals reveal the aqueous conditions associated with the alteration of carbonaceous chondrites, which is constrained by temperature, the water:rock (W/R) ratio, oxygen fugacity, and the solute composition[10]. Nakamura et al.[4] has reported a chemical equilibrium model for the aqueous alteration of the Ryugu host rock reacting with water containing $CO_2$, HCl, and organic matter that indicates a solution rich in magnesium (Mg), sodium (Na), and chloride (Cl) ions in the early stages of alteration[4].

Most of the constituents of CI chondrites are optically opaque matrices of secondary minerals[8] dominated by mixed phyllosilicates consisting of serpentine and saponite (81–84 vol%) as well as magnetite (6–10%), sulfides (4–7%), and carbonates (< 3%)[11]. Ryugu has been shown to be mineralogically similar to CI chondrites that have a pristine secondary mineral composition without terrestrial alteration[4,6] (see Table 1 for carbon abundances and isotopic compositions). A representative secondary mineral type that can be used for dating and reconstruction of solute compositions is carbonate[6,12]. To our knowledge, CI chondrites contain four types of carbonate minerals, dolomite [$CaMg(CO_3)_2$], breunnerite [$(Mg, Fe, Mn)CO_3$], calcite ($CaCO_3$), and siderite ($FeCO_3$), among which dolomite occurs most frequently[13], and references therein. Because carbonates can be dated by using the $^{53}Mn$-$^{53}Cr$ chronometer, which has a half-life of 3.7 Myr, they are valuable minerals for recording both the time when liquid water was present and the carbonate precipitation conditions[6,13–17]. Ryugu is important not only for these chemical reaction modeling results describing the secondary mineral precipitates, but also because it represents the pristine bulk Solar System composition[18,19] and contains a high diversity of primordial soluble organic molecules[5,20–25] and soluble inorganic ions[7]. However, how the major elements were redistributed during aqueous alteration by dissolution of primary minerals and precipitation of secondary minerals is still not fully understood. Direct snapshots of the chemical evolution of fluids during each dissolution and precipitation process are rare, except for observations of precipitate grains and desiccated veins[25,26].

Magnesium, which is one of the most ubiquitous elements in the secondary minerals of carbonaceous chondrites, exhibits large mass-dependent isotope fractionation during solid-liquid partitioning because of the large relative mass differences among $^{24}Mg$, $^{25}Mg$, and $^{26}Mg$[27,28]. Because carbonates and clay minerals are known to experience isotope fractionation at the time of their precipitation, the Mg isotope systematics of each partitioning process of the major carbonate and clay minerals are well calibrated[28–30]. The Mg isotopic composition ($\delta^{25}Mg$) of the originating fluid can therefore be reconstructed from that of the secondary minerals. One major difference among major secondary minerals is that the isotope fractionation factors ($\alpha$)

for carbonate ($\alpha < 1$) and clay ($\alpha > 1$) minerals are generally opposite in sign[28–30]. Because there are two species involved with opposite fractionation factors, the isotopic evolution of the residual liquid phase depends on (1) which phase precipitates first, (2) the values of the fractionation factors, and (3) the proportion of each precipitate. In addition, the ionic balance of the cation exchange pool of clay minerals can be used to unravel the composition of the latest contacted aqueous solution because clay minerals quickly achieve exchange equilibrium with the liquid phase[31]. In both experimental and natural samples, the Mg present in the exchange pool of phyllosilicates has been reported to show negligible Mg isotopic fractionation during adsorption and desorption with the ambient water[29,32]; thus, the exchangeable fraction is another proxy that records the isotopic composition of the dissolved Mg. The history of carbonaceous chondrites, such as thermal processing, mass transport, and mixing in the early Solar System, has been reconstructed from the element and isotopic composition of sequential acid extracts of carbonaceous chondrites[33,34]. To date, only one study has applied the sequential solvent extraction method to the exchangeable and acid-soluble phases of Ryugu samples to characterize the inorganic components[7] (Supplementary Fig. S1). However, the carbonate assemblage and exchangeable cations from the Ryugu sample constitute an inorganic proxy of the aqueous alteration of the bedrock and indigenous regolith of the carbonaceous asteroid.

In this study, we first report the Mg isotopic composition of breunnerite grains precipitated during the aqueous alteration of Ryugu and those of sequential solvent leachates, which provide information on the chemical evolution of fluids during the asteroid's history. Second, we identify the main component cations of two surface samples from Ryugu (A0106 from the first touchdown site, and C0107 from the second touchdown site; Fig. 1) that would have been partitioned their element compositions by aqueous alteration of the asteroid. We thereby obtain insights into the mode of precipitation of secondary minerals, the sources of dissolved cations, and the chemical evolution of fluids.

## Results and discussion
### Identification and chemical composition of breunnerite
Carbonate grains picked from Ryugu C0002 (Figs. 1, 2) were identified by laser Raman micro-spectroscopy to have spectrum peak (Fig. 2C) at 316.74 cm$^{-1}$, which is a specific feature of breunnerite (see also Supplementary Fig. S2). X-ray fluorescence spectroscopy (XRF) spectra for the same grain also showed that its main components were Mg and Fe, with a small amount of Mn. The positions of the main laser Raman microscopy peaks were then mapped spatially on the grain (Fig. 2D and E). Although rare peak perturbations originating from surface

**Table 1 | Summary of carbon (C, wt%) and nitrogen (N, wt%) profiles of Ryugu A0106 (i.e., first touchdown site) and C0107 (i.e., second touchdown site) and carbonaceous meteorites (Murchison, Murray, Tarda, Aguas Zarcas, Jbilet Winselwan) used for reference for assessing carbonate-bearing potentiality. Refer to the indicated references for detailed profiles of Ryugu[5,7,20], Tarda[61], Aguas Zarcas[62], Jbilet Winselwan[63], Murchison[5,25,64] and Murray[25,64]**

| Initial bulk sample | Weight | | Carbon | δ¹³C | Nitrogen | δ¹⁵N | C/N ratio | Reference |
|---|---|---|---|---|---|---|---|---|
| | µ gram (1σ) | Run | wt% (1σ) | ‰ VPDB (1σ) | wt% (1σ) | ‰ Air (1σ) | wt/wt (1σ) | |
| **Ryugu** | | | | | | | | |
| A0106 aggregate | 137.3 ± 40.9 | n = 3 | 3.76 ± 0.14 | −0.58 ± 2.0 | 0.16 ± 0.01 | 43.0 ± 9.0 | 23.5 ± 0.4 | Naraoka et al.[5] |
| C0107 aggregate | 126 ± 56.7 | n = 4 | 3.58 ± 0.47 | 1.22 ± 10.0 | 0.14 ± 0.01 | 36.8 ± 3.6 | 26.0 ± 2.4 | Oba et al.[20] |
| **Carbonaceous meteorite** | | | | | | | | |
| Murchison | 188 ± 27 | n = 3 | 1.78 ± 0.26 | −4.6 ± 0.8 | 0.10 ± 0.01 | 46.3 ± 1.0 | 18.9 ± 0.9 | This study |
| Murray | 133 ± 35 | n = 4 | 1.91 ± 0.07 | −3.5 ± 0.1 | 0.10 ± 0.01 | 67.2 ± 4.3 | 18.5 ± 1.1 | This study |
| Tarda | 173 ± 144 | n = 4 | 3.98 ± 0.16 | 13.9 ± 1.8 | 0.20 ± 0.01 | 57.9 ± 1.8 | 20.3 ± 0.5 | This study |
| Aguas Zarcas | 257 ± 79 | n = 4 | 1.70 ± 0.15 | −1.6 ± 2.0 | 0.07 ± 0.00 | 35.3 ± 0.9 | 23.4 ± 1.7 | This study |
| Jbilet Winselwan | 323 ± 177 | n = 4 | 1.47 ± 0.11 | −8.1 ± 0.4 | 0.08 ± 0.01 | 30.1 ± 1.9 | 17.8 ± 0.7 | This study |

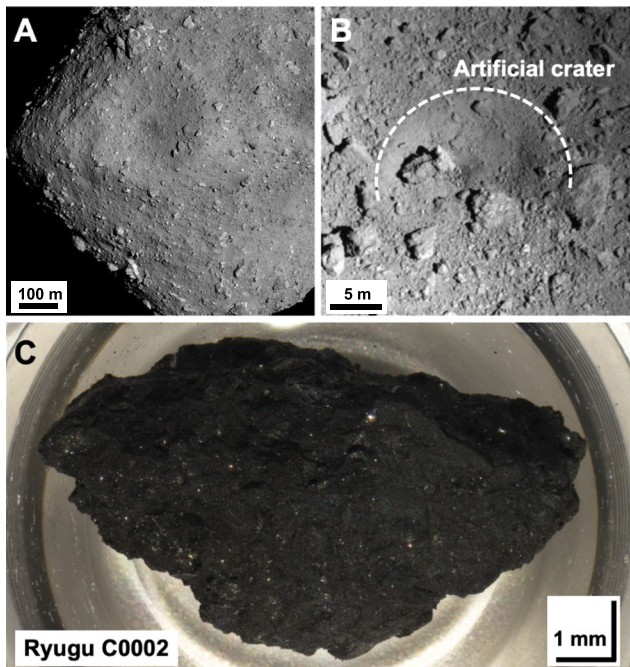

**Fig. 1 | The carbonaceous asteroid (162173) Ryugu and sub-surface sampling after the artificial crater formation. A** Photograph of Ryugu taken with the ONC-T camera onboard the Hayabusa2 spacecraft from 6 km on 20 July 2018 (Credit: JAXA, Univ. Tokyo, Kochi Univ., Rikkyo Univ., Nagoya Univ., Chiba Inst. Tech., Meiji Univ., Univ. Aizu, AIST). **B** Photograph was taken after the formation of the artificial crater by Small Carry-on Impactor operation[55]. The approximate size of the artificial crater is indicated by the dashed white line. **C** Photograph of Ryugu C0002 (total 93.5 mg) from the second touchdown sampling[1]. The photograph was taken under a microscope in the clean chamber of the JAXA curation facility before the sample distribution[2].

irregularities of the sample are observed, the positions of the breunnerite peak at the 316.74 $cm^{-1}$ (Fig. 2D) and positions of the carbonate peak (Fig. 2E) exhibit mostly homogeneous distributions; this result indicates that the grain is composed of a single mineralogy. The results

of elemental measurements by quadrupole inductively coupled plasma mass spectrometry (ICP-MS) after the microscopic analysis and dissolution of the grain in nitric acid (Table 2) showed that Fe/Mg was $0.319 \pm 0.021$ mol/mol. Other element compositions of breunnerite were Mn/Mg = $0.041 \pm 0.001$, Ca/Mg = $0.129 \pm 0.010$, and Na/Mg = $0.070 \pm 0.001$ mol/mol.

## Mg isotopic compositions

In Ryugu, the sample most enriched in heavy Mg isotopes was the HF/$HClO_4$ extract with a value of approximately −0.1‰ (Fig. 3), which constituted the residue of secondary mineral extraction. The sample most enriched in light Mg isotopes was the C0002 breunnerite with a value of −1.34‰ (Fig. 4). In the other extracts, the $\delta^{25}Mg$ values were intermediate; the A0106 $H_2O$ extract had a low $\delta^{25}Mg$ value of −0.85‰, but the $\delta^{25}Mg$ value in the C0107 $H_2O$ extract was similar to that of the exchangeable form in the $NH_4Cl$ extract. In the ethylenediaminete-traacetic acid (EDTA) extracts of both A0106 and C0107, $\delta^{25}Mg$ was around −0.7‰, whereas in the $CH_3COOH$ extract, it was slightly higher at −0.5‰.

The $\delta^{25}Mg$ value in the A0106 $H_2O$ extract (Fig. 3, Table 2) was comparable to those in the hot $H_2O$ extracts obtained during sequential soluble organic matter (SOM) leaching (−0.81‰ and −0.80‰; Fig. 3, Table 2). The $\delta^{25}Mg$ values in the EDTA and $CH_3COOH$ extracts of this sample are clearly lower than that of the HCOOH extract (−0.3‰). The main target phase of the EDTA, $CH_3COOH$, and HCOOH extractions is carbonate, whereas in the HCl and HF/$HClO_4$ extracts for silicate minerals, $\delta^{25}Mg$ values were −0.1‰ to −0.2‰, values close to that of bulk chondrites (Fig. 3). The $\delta^{26}Mg*$ value, the $^{26}Mg$ excess resulting from $^{26}Al$ decay, in each leachate in this study and in the SOM leachates reported by Yoshimura et al.[7] are indistinguishable from 0‰ within analytical uncertainty (Supplementary Fig. S3).

The EDTA fraction targets carbonates that are more soluble in acid than in $CH_3COOH$ (e.g., the former is used for selective extraction of calcite, and the latter for selective extraction of dolomite, see Supplementary Fig. S1). The plot of Ca/Mg versus $\delta^{25}Mg$ (Fig. 5) suggests that there is a contribution from the breunnerite component in the EDTA fraction. In contrast, the $CH_3COOH$ fraction plots on a mixing line between the total digests of the Ryugu samples and dolomite; this

**Table 2 | Normalized Mg isotopic ratios $\delta^{26}Mg$, $\delta^{25}Mg$, and $\delta^{26}Mg*$ with 2 SD errors (vs. the DSM-3 standard) and Mg/Ca, Mg/Fe, Mg/Al, and Mg/(Mg+Fe) element ratios for each extracted fraction and breunnerite**

| Sample | $\delta^{26}Mg$ | 2 SD | $\delta^{25}Mg$ | 2 SD | $\delta^{26}Mg*$ | 2 SD | Mg/Ca (mol/mol) | Mg/Fe (mol/mol) | Mg/Al (mol/mol) | Mg/Mg+Fe (mol/mol)*100 |
|---|---|---|---|---|---|---|---|---|---|---|
| **C0002** Breunnerite | −2.60 | 0.04 | −1.34 | 0.02 | 0.018 | 0.016 | 7.68 | 3.14 | - | 75.8 |
| **A0106** | | | | | | | | | | |
| $H_2O$ | −1.59 | 0.05 | −0.85 | 0.02 | 0.061 | 0.053 | 9.66 | 4.56 | 4.82 | 82.0 |
| $NH_4Cl$ | −1.19 | 0.09 | −0.67 | 0.09 | 0.113 | 0.125 | 8.89 | 8.31 | 36.41 | 86.2 |
| EDTA | −1.33 | 0.11 | −0.67 | 0.06 | −0.017 | 0.042 | 9.91 | 2.47 | 25.63 | 70.0 |
| $CH_3COOH$ | −1.00 | 0.05 | −0.50 | 0.04 | −0.029 | 0.051 | 2.98 | 1.71 | 15.41 | 63.5 |
| HF+$HClO_4$ | −0.23 | 0.02 | −0.10 | 0.04 | −0.034 | 0.046 | - | 1.15 | 11.80 | 53.4 |
| #7-1 hot $H_2O$ (SOM) | −1.54 | 0.03 | −0.81 | <0.01 | 0.054 | 0.045 | 1.82 | 117.98 | - | 99.2 |
| **C0107** | | | | | | | | | | |
| $H_2O$ | −1.35 | 0.04 | −0.68 | 0.04 | −0.025 | 0.051 | 2.89 | 4.58 | 16.52 | 82.1 |
| $NH_4Cl$ | −1.19 | 0.05 | −0.61 | 0.04 | 0.009 | 0.006 | 4.00 | 161.50 | 649.68 | 97.0 |
| EDTA | −1.33 | 0.05 | −0.69 | 0.04 | 0.024 | 0.011 | 3.46 | 2.44 | 34.29 | 67.3 |
| $CH_3COOH$ | −0.97 | 0.06 | −0.50 | 0.05 | 0.005 | 0.008 | 3.24 | 1.20 | 14.84 | 56.4 |
| HF+$HClO_4$ | −0.17 | 0.06 | −0.07 | 0.03 | −0.035 | 0.046 | - | 1.25 | 11.65 | 55.5 |
| #7-1 hot $H_2O$ (SOM) | −1.46 | 0.09 | −0.80 | 0.06 | 0.106 | 0.111 | 3.99 | 93.80 | - | 98.9 |
| **Serpentine** | | | | | | | | | | |
| $CH_3COOH$ | −1.81 | 0.09 | −0.94 | <0.01 | 0.038 | 0.022 | - | 6.78 | 11.79 | 87.2 |
| HF+$HClO_4$ | −0.24 | 0.07 | −0.09 | 0.02 | −0.064 | 0.084 | - | 13.22 | 31.51 | 93.0 |
| Cambridge 1 (n = 9) | −2.61 | 0.05 | −1.35 | 0.04 | 0.036 | 0.045 | | | | |

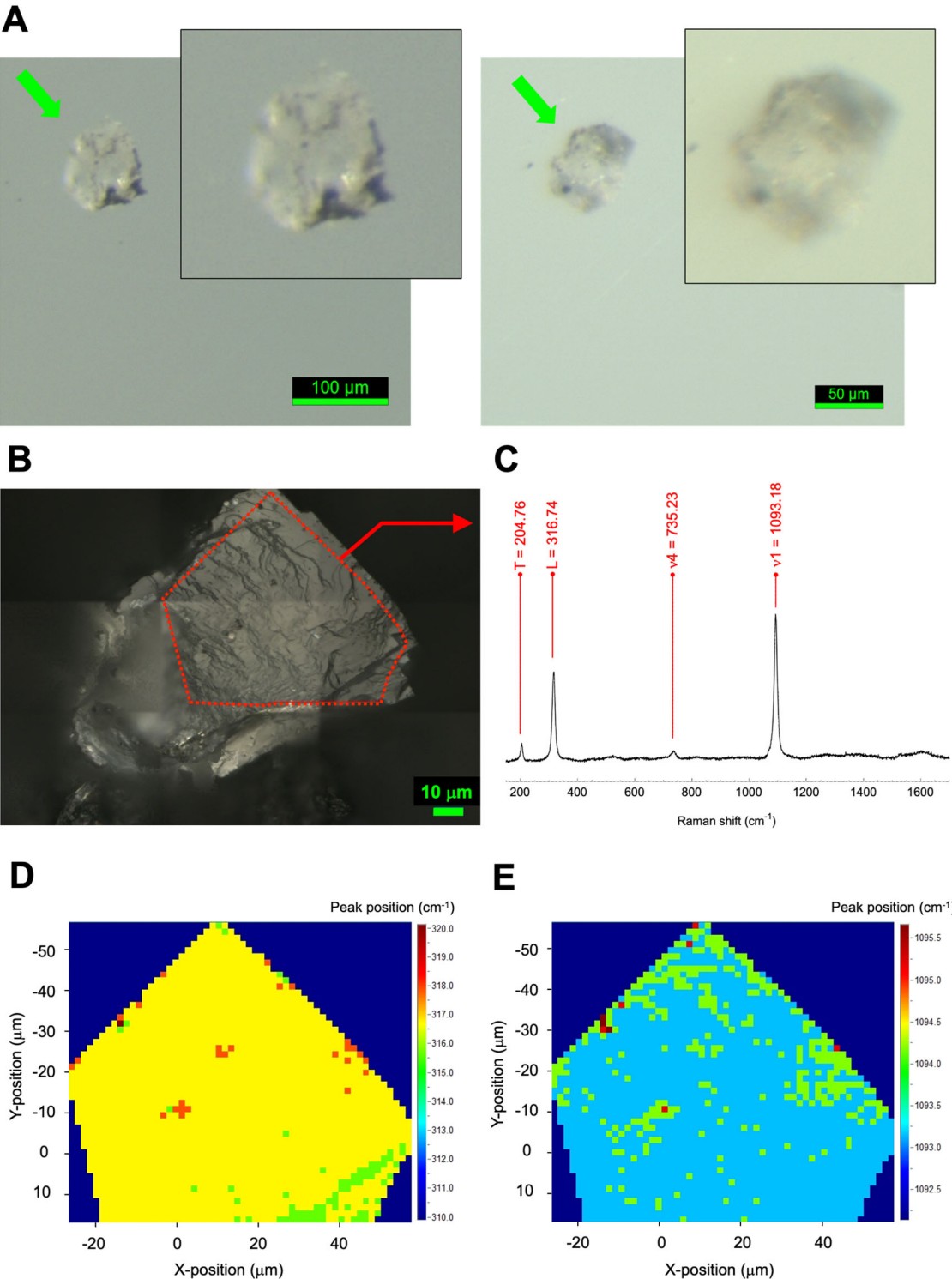

**Fig. 2 | Identification of a breunnerite grain from the asteroid (162173) Ryugu.**
**A** Micrographs of the two largest single-mineralogy breunnerite grains isolated from Ryugu C0002 and **B** the region of interest (ROI; within the dotted red line frame) of the larger of the two grains for non-destructive spectroscopy. The two breunnerite grains were first assessed by non-destructive analysis. Then, a high-precision analysis of the magnesium ($^{24}$Mg, $^{25}$Mg, $^{26}$Mg) isotope systematics of the two grains was carried out. **C** Raman shift signatures of the ROI analyzed by the laser Raman microscopy (method after Urashima et al.)[54] and spatial mapping of the **D** breunnerite-specific peak (316.74 cm$^{-1}$) and the **E** CO$_3$-specific peak (1093.18 cm$^{-1}$). The horizontal (X-position, μm) and vertical (Y-position, μm) positions in the ROI are shown at pixel resolution.

result suggests a significant contribution mainly from dolomite to the CH$_3$COOH fraction. However, because CH$_3$COOH also partially dissolves phyllosilicates (Supplementary Table S1), Mg derived from phyllosilicates was also present in that fraction (Fig. 5). The HCl fractions are characterized by higher δ$^{25}$Mg and lower Ca/Mg compared

with the four bulk digests of Ryugu samples[35]. The compositions of the sequential extracts with carbonate (dolomite and breunnerite) and phyllosilicate are used as mixing endmembers of Mg. The HF/HClO$_4$ fractions are not shown on the Ca/Mg plot because Ca was at the lower limit of detection.

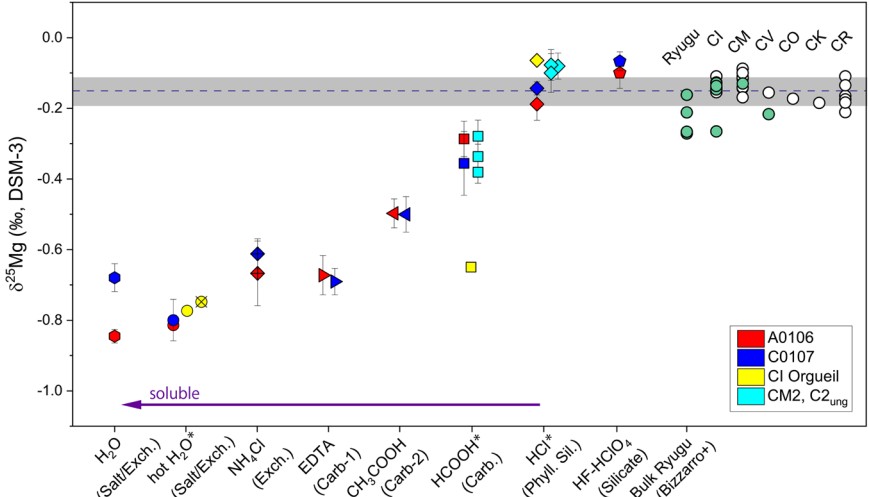

**Fig. 3 | Magnesium isotope profiles of the sequential extracts (H₂O to HF-HClO₄).** Chemical fractions obtained in this study and in the SOM sequential leaching with 2 SD errors (hot H₂O, HCOOH, and HCl data are from Yoshimura et al.[7]; method after Naraoka et al.)[5,7]. All data for HCOOH and HCl marked by asterisks and a part of the hot H₂O fraction (only Orgueil) have been published in Yoshimura et al.[7]. The yellow circle with the cross represents the data from the #5 H₂O extract of CI-group Orgueil; these data are consistent with the #7 hot H₂O data (Supplementary Fig. S1). Exch. means exchangeable pool. The horizontal dashed line and gray band show the average of bulk values for the reference meteorites and the 2σ errors[37]. The horizontal arrow indicates the general trend toward more soluble components; Supplementary Fig. S1 shows the minerals that each solvent targeted for extraction. Open circles represent high-precision δ²⁵Mg reference data for carbonaceous chondrites (CC)[38–41], and the blue-green circles represent the bulk data of four Ryugu grains and CI (Orgueil), CM, and CV meteorites reported by Bizzarro et al.[35]. For many of these reference CC data (CI, CM, CV, CO, CK, CR) as well as for the bulk Ryugu grains, the 2SE measurement error is within the size of the symbol.

## Chemical composition of solute extracts from aggregate samples

The hot H₂O extracts of A0106 and C0107 contained the highest amount of Na, followed by Mg. The amount of Ca varied greatly between the samples; C0107 contained 3.8 times more Ca than A0106 when normalized by sample weight, and 3.4 times more Ca when normalized by Mg. Thus, Ca was the most heterogeneous of all measured elements. The molar ratios of Mg, Ca, and Na+K to their sum are shown in a ternary diagram (Fig. 6, Supplementary Table S2), on which the contributions of Na+K and Mg range from 55% to 35%. The NH₄Cl extracts were characteristically even more enriched in Na and K than the H₂O extract. Alkali metals alone accounted for about 80% of the major cations among the exchangeable cations. This composition is similar to the hot H₂O extract in the sequential SOM leaching[7].

The ternary diagram approach for examination of element partitioning was not applied to the EDTA, CH₃COOH, and HF/HClO₄ extracts because Na contaminated all of the solutions after the first leaching for carbonates with the solution prepared from EDTA disodium salt. This contamination is a necessary trade-off when carbonates are extracted with weak acids. Note, however, that the SOM leaching results show that more than 90% of the alkali metals leach out in hot H₂O[7]. Furthermore, Mg/(Mg+Fe) was low in the EDTA, CH₃COOH, and HF/HClO₄ extracts compared to that in the H₂O and NH₄Cl extracts and below 70 after EDTA extraction, indicating that the dissolution of Fe-containing mineral phases had increased (Table 2).

The Mg/Fe value of 2.44–2.47 mol/mol in EDTA extracts is relatively close to its value of 3.14 mol/mol in the C0002 breunnerite (Table 2). In contrast, Mg/Fe of the CH₃COOH extracts targeting dolomite is 1.20–1.71 mol/mol, significantly lower than that of the breunnerite; this carbonate dissolution behavior is consistent with the Ca/Mg and δ²⁵Mg results (Fig. 5). The Mg/Fe value of the HCOOH extract in the SOM leaching (~2.4 mol/mol) is also close to the EDTA value, and the Mg/(Mg+Fe) of the EDTA extract of ~70 mol/mol is close to that of the HCl extract of SOM leaching, which contains the phyllosilicate fraction (Table 2). In the ternary diagram (Fig. 6), the final residue after HF/HClO₄ extraction is similar in composition to the Solar System abundances and the bulk composition of CI meteorites, as well as to the HCl extracts in the SOM leaching.

## ²⁵Mg/²⁴Mg homogeneity of bulk carbonaceous chondrites and mass-dependent Mg isotope fractionation in carbonates

The abundances and isotopic compositions of C, N, and H using tens to hundreds of μg of the A0106 and C0107 samples used in this study are similar to previous analyses of CI chondrites from Ivuna and Orgueil (Table 1). Therefore, the influence of lithological heterogeneity on the representativeness of the chemically extracted values for these light elements is expected to be small. Other values measured for CM2 Murchison, Murray, Aguas Zarcas, CI Orgueil, and C2_ung Tarda at similar weight scales are shown for comparison of chemical composition. However, there is a variation of ~0.1‰ in the Mg isotopic composition of Ryugu[35]. Such variation is explained by the varying amount of carbonates in each aliquot[18], indicating the effect of taking only small aliquots.

Previous studies have reported relatively homogeneous Mg isotopic compositions of δ²⁵Mg = −0.15 ± 0.04‰ (2σ) and δ²⁶Mg = −0.28 ± 0.06‰ (2σ) for carbonaceous, ordinary, and enstatite chondrites, including the nine chondrite groups (CI, CM, CO, CV, L, LL, H, EH, and EL)[36,37]. The compositional homogeneity of chondrites indicates that at the scale of bulk samples, the Mg isotopic compositions of their initial materials were similar and that mass-dependent isotope fractionation (δ²⁵Mg and δ²⁶Mg) did not occur at the per mil level (Fig. 3). δ²⁶Mg is affected by the radioactive decay of ²⁶Al. However, the deviation should be on the order of tens of ppm and thus generally well below the mass-dependent variation observed on the 0.1 permil level. To eliminate potential radioactive decay effects, δ²⁵Mg, which is not affected by ²⁶Al decay, is used to examine Mg partitioning in the aqueous alteration of Ryugu. Recently, small but resolvable ²⁵Mg/²⁴Mg changes in bulk chondrites have been reported by high-precision Mg isotope analyses[35,38–41]. However, the maximum extent of the variation is about 0.1‰, which is sufficiently small compared with the magnitude of the δ²⁵Mg variation observed in leachates and the single-mineralogy grains of this study.

Bulk δ²⁵Mg values of samples returned from Ryugu has been reported by Bizzarro et al.[35] (Fig. 3). The δ²⁵Mg values of four samples

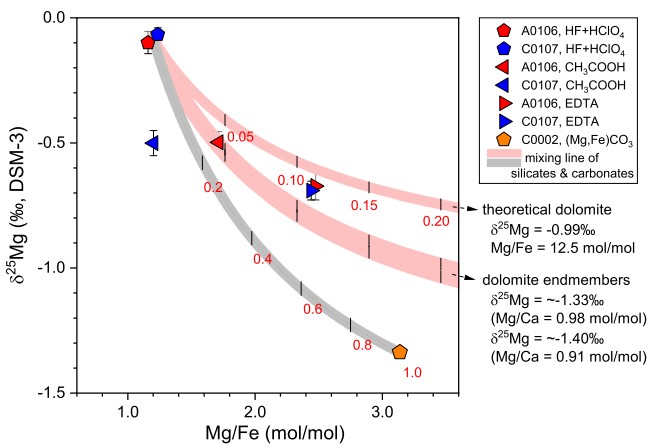

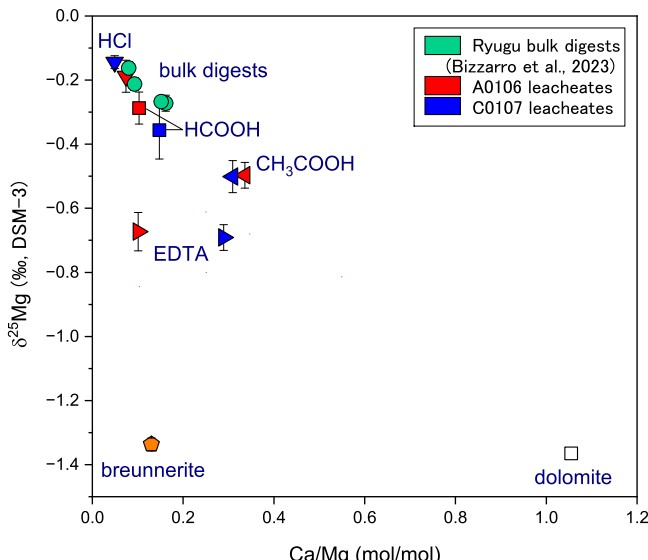

**Fig. 4 | Magnesium isotope systematics of the breunnerite grains (C0002) and leachates extracted from the aggregated samples (A0106 & C0107) and dolomite endmembers of Ryugu.** The $\delta^{25}$Mg isotopic ratio (‰, vs. the DSM-3 standard with 2 SD errors) versus Mg/Fe (mol/mol) of the Ryugu C0002 breunnerite single-mineralogy grains. The raw profiles of the carbonate leaching fractions (EDTA and $CH_3COOH$ fractions) and the residual fraction (HF/HClO$_4$) were compiled using the same formulation. Mixing curves of the calculated dolomite end component with the residual fraction and the mixing curve of the breunnerite with the residual fraction are shown in red and gray, respectively. Theoretical dolomite, which denotes the $\delta^{25}$Mg value assuming direct uptake of the $Mg^{2+}$ resulting from the dissolution of the primary minerals, was calculated using the bulk composition of the C0108 sample ($\delta^{25}$Mg = −0.16‰)[35] as the initial solution composition, the isotope fractionation factor[42], and a precipitation temperature of 37 °C[6]. The theoretical Fe/Mg ratio of 12.5 mol/mol is the average value of the Ryugu samples[4,12,43,65] (Supplementary Table S3). The Mg isotope ratio of the dolomite precipitated on Ryugu was calculated by Bizzarro et al.[35] ($\delta^{25}$Mg = −1.40‰, Mg/Ca = 0.91)[35,43]. Because the calculation method is affected by the Mg/Ca ratio of the dolomite, another potential endmember ($\delta^{25}$Mg = −1.33‰, Mg/Ca = 0.98) obtained by applying the Mg/Ca value of Fujiya et al.[12] is also shown. The Mg/Ca values reported by Nakato et al.[65] and Nakamura et al.[4] (not shown) are intermediate between those of Bazi et al.[43] and Fujiya et al.[12] (Supplementary Table S3). The effect of partial dissolution of labile phyllosilicates is found even in the EDTA extract (for selective dissolution of carbonates) resulting in low Mg/Fe and slightly high $\delta^{25}$Mg. The $CH_3COOH$ extracts show a particularly high contribution from phyllosilicates because of their greater acidity (Supplementary Table S1).

**Fig. 5 | Mg isotope ratios with 2 SD errors and Ca/Mg molar ratios in the carbonate and phyllosilicate leaching fractions, breunnerite, and dolomite.** Values for bulk digests of the Ryugu aggregate samples were reported by Bizzarro et al.[35]. Data for the HCl and HCOOH fractions are from Yoshimura et al.[7] (Supplementary Table S5). The HF/HClO$_4$ fraction of this study is not shown because Ca is below the detection limit.

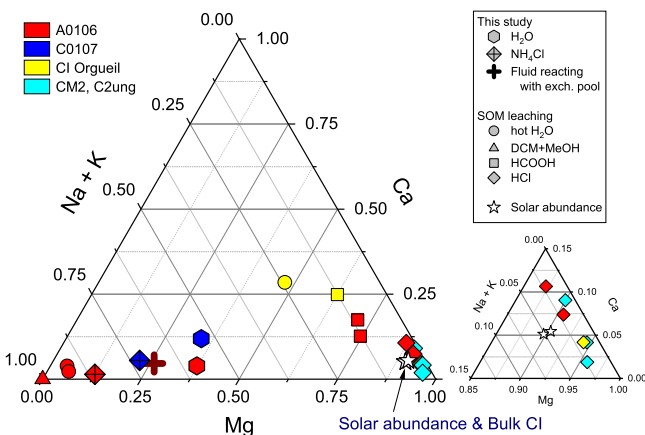

**Fig. 6 | Ternary diagram showing the molar proportions of Mg, Ca, and Na + K in leachates extracted from the Ryugu aggregated samples (A0106 & C0107).** The molar proportions in the sequential extracts of this study and in the SOM leaching of Ryugu samples A0106 (red) and C0107 (blue), Orgueil (yellow), and Tarda, Aguas Zarcas, and Jbilet Winselwan (light blue), with the bulk compositions of CI chondrite and Solar System abundance[66] (stars; compiled after Lodders, 2021; an enlarged plot is at the bottom right) also plotted for reference. The types of solvent and the main target phases of the leaching experiments are documented in Supplementary Fig. S1. The wine-colored cross indicates the chemical composition of the fluid in contact with the cation exchange pool of Ryugu, calculated using the partitioning coefficients for the major cations[31].

ranged from −0.16 ± 0.02‰ to −0.27 ± 0.03‰; thus, they are slightly lighter than most literature data of Ivuna-type (CI) and other carbonaceous chondrite groups. Most of the Orgueil data reported by Bizzarro et al.[35] also agree with the literature carbonaceous chondrite values, although one measurement is nearly identical to the lowest value of Ryugu[35]. This variation in $\delta^{25}$Mg is attributable to the small sample size and is interpreted to reflect heterogeneous sampling of isotopically lighter carbonates formed by low-temperature equilibrium processes with an endmember composition of about −1.40‰[35]. The bulk $\delta^{25}$Mg value of C0108, in which the carbonate influence is small, is indistinguishable from that of bulk CI chondrites. Here, the C0108 value of −0.16‰ is adopted as the $\delta^{25}$Mg value of $Mg^{2+}$ in the initial fluid resulting from the dissolution of primary minerals during aqueous alteration, as will be shown later.

In general, solutes extracted by chemically mild solvents are considered to have higher solubility and therefore to precipitate during later aqueous alteration stages. It should be noted that the precipitation of minerals would have proceeded in the opposite order to the extraction step sequence (Supplementary Fig. S1). The overall trend of our $\delta^{25}$Mg composition results is that the more soluble components have lower values and, as the acidity of the extraction fluid increases (i.e., for extraction of phyllosilicates and residues), $\delta^{25}$Mg becomes closer to the bulk composition (Fig. 3). Furthermore, we successfully analyzed grains of breunnerite with a $\delta^{25}$Mg value of

−1.34 ± 0.02‰ (Fig. 4). This value was obtained by dissolving the two grains shown in Fig. 2A together. Because of the high reactivity of Ryugu's phyllosilicates, it is difficult to obtain carbonate $\delta^{25}$Mg by chemical extraction because even EDTA causes partial dissolution, so direct measurement of the isotope ratio of the microparticles is an effective technique. By considering isotopic fractionation factors during Mg-bearing carbonate precipitation, the process by which dissolved Mg is removed from the fluid can be reconstructed, as discussed later. Note that the timing of precipitation can vary depending

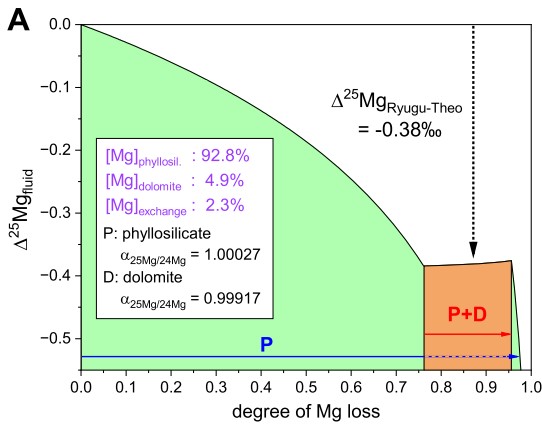

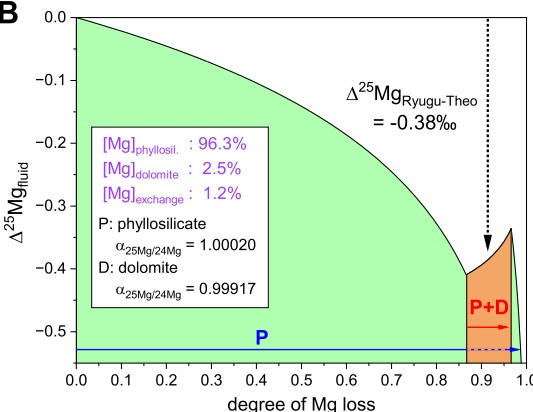

**Fig. 7 | Model of Mg isotopic changes during aqueous alteration on Ryugu.** In this model, the initial solution was set to zero Mg loss and $\Delta^{25}Mg_{fluid}$ = 0‰, and dissolved Mg was assumed to be partitioned into two dominant Mg host phases, phyllosilicate and dolomite. This model accounts for two isotopic tie points: (1) the offset of $\delta^{25}Mg$ values between theoretical and Ryugu dolomite ($\Delta^{25}Mg_{Ryugu-Theo}$ = −0.38‰)[35], shown in Fig. 4, and (2) the isotopic composition of dissolved Mg in the final stage that liquid $H_2O$ was present ($\Delta^{25}Mg_{fluid}$ = −0.55‰ compared to the initial solution) estimated from the cation exchange pool of phyllosilicates (Supplementary Table S4). The former reflects the timing of the dolomite precipitation event, and the latter reflects the composition before the complete loss of liquid $H_2O$. The $^{25}Mg/^{24}Mg$ fractionation factor α for dolomite is from Li et al.[42], and its formation at 37 ± 10 °C[6] was calculated. The α values for phyllosilicates were calculated by assuming isotopic fractionation in a closed system. The Mg contents in the residual fluid is present at a 5% relative to dolomite (see Supplementary Information), with the Mg isotopic composition of −0.55‰ (this study). Model results based on elemental mass balance of Ryugu A0106 and C0107, for (**A**) a Mg abundance ratio of 92.8:4.9:2.3 for phyllosilicates, dolomite and exchangeable fractions and (**B**) for 96.3:2.5:1.2 (see Supplement Fig. S6 for sensitivity experiments). In the orange sections, Mg removal to dolomite and phyllosilicate occurs simultaneously. The Mg removal rate to dolomite was assumed to be one-third that of phyllosilicate in order to achieve conditions of low isotopic ratio heterogeneity in dolomite; cases with varying precipitation rates are shown in Supplementary Fig. S6. We used the following equation for closed-system fractionation: $\Delta^{25}Mg_{fluid} = \delta^{25}Mg_0 + 1000(\alpha-1)\ln(f_w)$ where $\Delta^{25}Mg_{fluid}$ and $\delta^{25}Mg_0$ are the Mg isotopic composition of the fluid and its initial value (set at 0); α is the fractionation factor between the dissolved Mg and secondary minerals; and $f_w$ is the fraction of Mg remaining in the fluid.

on the Mg isotopic fractionation factor during dolomite precipitation and the partitioning ratio of Mg from solution to dolomite and phyllosilicate (see Supplementary Information). The Mg in Ryugu's secondary minerals may have been derived mainly from Mg-rich olivine and pyroxene. Aqueous alteration of the Ryugu parent body is thought to have consisted of reactions between rocks and $H_2O$ with $CO_2$ and HCl[4]. Accretion of HCl-containing ice and the subsequent dissolution of primary minerals results in an initial fluid composition rich in Mg and Fe, followed by neutralization of the solution, and serpentine, saponite, and carbonate precipitation, thereby transforming it into a Na-rich alkaline fluid[10]. The precipitation order, including the pre- and post-sequence and the simultaneous precipitation of these secondary minerals, is of interest for understanding the mode of supply and the movement of solutes into and their removal from the fluid.

## Insights from $\Delta^{25}Mg$ profiles and in situ temperature
To determine the $\delta^{25}Mg$ value of the calcifying fluids, we sought to reconstruct the fluid chemistry using the $\Delta^{25}Mg$ of dolomite as follows:

$$\Delta^{25}Mg_{dolomite-aq} = \delta^{25}Mg_{dolomite} - \delta^{25}Mg_{aq} \approx 10^3 \ln \alpha \qquad (1)$$

where $\delta^{25}Mg_{dolomite}$ and $\delta^{25}Mg_{aq}$ are the Mg isotopic composition of the dolomite and aqueous solution, respectively, and α is the fractionation factor between the dissolved Mg and dolomite. Carbonates obtained under natural conditions or by inorganic precipitation are enriched in $^{24}Mg$ relative to the solution[28]. The most common carbonate found in Ryugu as well as in CI chondrites is dolomite[4]. The Mg extracted in the HCOOH extract averaged between A0106 and C0107 was ~1160 μmol per gram of initial solid and was derived mainly from carbonates. By assuming that the Mg extract was derived from dissolved carbonates and a W/R ratio of 0.2–0.9 during the alteration[4], the Mg concentration of the solution was estimated to be 1–5 mol/L. In this study, $\Delta^{25}Mg_{dolomite-aq}$ was calculated according to the fractionation equation of Li et al.[42] based on the similarity of their precipitation conditions (1 M $MgCl_2$ and 1 M $CaCl_2$). Given a dolomite precipitation temperature on Ryugu of 37 ± 10 °C[6], $\Delta^{25}Mg_{dolomite-aq}$ was calculated to be −0.83 ± 0.11‰.

The Mg isotope ratios of the carbonates provide insight into the partitioning of the major constituent element Mg and precipitating conditions. The main factors controlling the isotope fractionation of carbonates are temperature and precipitation rate, and calibrations of these factors have been reported for each carbonate polymorph. However, in the case of inorganic carbonates precipitated from solutions with relatively high Mg/Ca ratios ( > ~3 mol/mol) and saturation indices (Ω > ~3; e.g., $\Omega_{calcite} = [Ca^{2+}] [CO_3^{2-}]/K_{sp}$, where $K_{sp}$ is the stoichiometric solubility product of calcite), the precipitation rate has no effect on mass-dependent $^{25}Mg/^{24}Mg$ fractionation and the temperature dependency is ~0.005‰/°C[28]. When the 2σ value of the external repeatability of $\delta^{25}Mg$ is about ± 0.05‰[36], $\delta^{25}Mg$ may be a temperature indicator with an error of ~10 °C, if it follows an isotope equilibrium reaction. Among the carbonates, the fact that calcite precipitated earlier suggests that dolomite, in contrast, formed during retrograde cooling when the fluid and silicate approached oxygen isotopic equilibrium[16]. Because the breunnerite may have formed at a later stage of aqueous alteration than dolomite, it may be possible to use $\delta^{25}Mg$ to detect the difference in precipitation temperature.

## Precipitation order of Mg-bearing secondary minerals
The $\delta^{25}Mg$ of the dolomite endmember was calculated to be −1.40‰ to −1.33‰ (Fig. 4) from the linear relationship between $\delta^{25}Mg$ and Mg/Ca of the bulk samples[35] and Mg/Ca of dolomite[12,43] (summarized in Supplementary Table S3). These values are 0.34–0.41‰ lower than the isotopic composition of dolomite, which is assumed to have precipitated directly from fluid containing $Mg^{2+}$ supplied by primary silicate mineral dissolution at the very beginning of aqueous alteration (Fig. 4). This difference in $\delta^{25}Mg$ can be considered evidence of Mg uptake by secondary phyllosilicates prior to the precipitation of the dolomite. In general, the preferential uptake of $^{25}Mg$ rather than $^{24}Mg$ by phyllosilicates during their formation results in a $\delta^{25}Mg$ decrease in the residual solution. In addition, the $\delta^{25}Mg$ value of the exchangeable

pool indicates that the decrease of the isotopic composition from that of the initial bulk Ryugu is −0.55‰ (see also the next section), which can be considered as the endpoint of Mg isotopic changes in aqueous alteration.

A model for the precipitation of phyllosilicates and dolomite in a closed system has been constructed with a Mg partitioning ratio of up to 90:10 for phyllosilicates and dolomite[35], the main Mg-host phases in Ryugu. Furthermore, the observed isotopic composition of dolomite and exchangeable forms at Ryugu constrain the fractionation factor for Mg structurally incorporated into phyllosilicates in a closed system. If an isotope fractionation factor of 0.99917[42] is applied to Ryugu dolomite, a simple model for Mg partitioning during aqueous alteration on Ryugu indicates that the $Mg^{2+}$ is first precipitated in phyllosilicates with a $^{25}Mg/^{24}Mg$ fractionation factor of 1.00027 (Fig. 7A). The solution from which dolomite would precipitate afterward would have a composition corresponding to the observed −0.38‰ decrease of $\delta^{25}Mg$ (Fig. 7A). Secondary phyllosilicate precipitation continued after dolomite precipitation, and the final $\delta^{25}Mg$ composition of the solution was recorded in the cation exchange pool to have decreased further by −0.55‰ compared to the $\delta^{25}Mg$ composition of the initial solution (Fig. 7A, Supplementary Table S4). This precipitation order is consistent with the chemical reaction modeling results for Ryugu[4], which show that saponite precipitates over a lower W/R range, corresponding to low-alteration lithologies without dolomite. Thus, the $\delta^{25}Mg$ compositions allow quantification of the Mg partitioning history.

When a Mg partition of 2.5% for dolomite is applied instead, dolomite precipitation occurs at a later stage of aqueous alteration, and the fractionation factor of phyllosilicate is calculated to be 1.00020 (Fig. 7B). Under both Mg partitioning conditions (Fig. 7A, B), removal of $Mg^{2+}$ by phyllosilicate precedes the precipitation of dolomite, and the timing of dolomite precipitation is calculated to commence when the residual $Mg^{2+}$ is approximately 76% to 87 % of the original budget. In the future, it will be necessary to determine bulk representative values of Mg abundances of dolomite and phyllosilicate at Ryugu, and to comprehensively determine the crystallization sequence during aqueous alteration based on oxygen and carbon isotope ratios of dolomite and age values. Oxygen and carbon isotope ratios of the Ryugu dolomite indicate that it was formed during retrograde cooling in the late stages of aqueous alteration[12,44]. From the viewpoint of Mg partitioning, it is also supported that dolomite precipitation occurred at a time when partitioning was well advanced.

## Partitioning of $Mg^{2+}$ in cation exchange pools and solute compositions

In both A0106 and C0107, the most abundant alkali metals in the $H_2O$ extracts were Na and K, together accounting for about 56% of the total budget; Mg was next with 36%, and Ca was the least abundant. Clay minerals have two metal reservoirs with different chemical behaviors and Mg isotopic ratios: exchangeable Mg and structurally bound $Mg^{2+}$[29]. The lowest $\delta^{25}Mg$ value of the $H_2O$ extracts was about −0.8‰ in both this study and in the SOM leaching[7]; but in contrast, the $\delta^{25}Mg$ value of the clay mineral fraction in the SOM leaching was −0.17‰ and that in the residual fraction ($HF/HClO_4$ extract) of the present study was −0.08‰. The main Mg host in clay minerals and in the residual phyllosilicate fractions is probably structural Mg, which has higher $\delta^{25}Mg$ than the olivine average[45]. The identification of mineral phases at micro- to nanoscale also rules out the presence of secondary minerals composed of inorganic Mg salts[4]. From the cation exchange pool of phyllosilicates (e.g., surface sites and exchangeable interlayer sites), isotopic tracers have shown that dissolved Mg is taken up without fractionation[29,32]. The $\delta^{25}Mg$ of the $NH_4Cl$ fraction extracted after the $H_2O$ extraction is −0.64‰, which is 0.5‰ lower than that of the clay fraction in the SOM leaching (i.e., HCl extracts, Fig. 3). The exchangeable ions equilibrate to changes in solution composition through rapid exchange reactions[31]. Therefore, the solute composition

of the exchange pool should be close to the solute composition at the final stage of aqueous alteration when liquid $H_2O$ was still present on Ryugu.

## Ionic composition of the fluid in contact with cation exchange pools

The $H_2O$ extracts described above have a lower proportion of alkali metals than the hot $H_2O$ extract in the SOM leaching, and it is the exchangeable fractions that explain this difference: in A0106 and C0107, 86% and 73%, respectively, of the exchangeable fraction consists of Na and K, reflecting the selectivity of the cation exchange pool of saponite for alkali metals and the aqueous alteration solution composition. Cation exchange reactions of clay minerals are reversible, and the general ion selectivity for monovalent and divalent cations increases in the following order[46]:

$$Li^+ < Na^+ < K^+ < Rb^+ < Cs^+ \text{ within the alkaline metals group(1A)},$$

$$Mg^{2+} < Ca^{2+} \leq Sr^{2+} \leq Ba^{2+} \text{ within the alkaline earth metals group(2A)}.$$

Here, potassium and magnesium are known to have comparable ion selectivity when monovalent and divalent ions are compared[46]. As is usual for strongly acidic ion exchangers, common clay minerals (e.g., kaolinite and montmorillonite) show this selective adsorption of $Na^+$ before $Li^+$, but saponite is more selective for $Li^+$ than $Na^+$[47]:

$$Na^+ < Li^+ < K^+ < Rb^+ < Cs^+$$

Cation exchange is a rapid chemical reaction between the cations of the dissolved phase and the mineral surface (especially clay minerals), wherein a cation exchange pool is formed on the reactant surface and the negative charge on the clay surface is balanced[31]. Because we eluted exchangeable cations separately, first with $H_2O$ and then with $NH_4Cl$, rather than directly with $NH_4Cl$, the total cations extracted during these steps are merged. Because the elemental ratios in this exchangeable pool were considered to be in exchange equilibrium with the ambient dissolved ions until the latest stage of aqueous alteration, the chemical composition of the fluid was reconstructed from the partition coefficients of the cations.

To reconstruct the dissolved ion composition, we applied partition coefficients of $K_{Na/Mg} = 0.5$, $K_{Na/Ca} = 0.4$, and $K_{Na/K} = 0.2$ between the fluid and cation exchange pool[31] and the average element ratios in the exchange pool. Owing to the presence of potential heterogeneity in the fluid composition corresponding to differences in the degree of alteration and W/R, calculations were carried out using the averaged values for A0106 and C0107. The Na/Mg, Na/K, and Na/Ca values of the Ryugu exchange pool were calculated to be 0.97, 0.49, and 4.3, respectively. Given the above partition coefficients, the chemical composition of the cation exchange pool demonstrates that Na was the most abundant dissolved cation in the fluid in contact with the cation exchange pool of the clay minerals with molar ratios of $Na^+$ to $Mg^{2+}$, $K^+$, and $Ca^{2+}$ of 1.9, 2.5, and 10.7, respectively (Fig. 6). The ratio of the Na concentration to the K concentration ranges from 1.3 to 2.1, when $K_{Na/K}$ values of 0.24 to 0.38 obtained from the laboratory experiments with saponite are applied[47]. Whether natural or experimental ideal substitution reactions reflect actual substitution reactions should be investigated in the future (e.g., ionic composition, effects of ionic strength, and pH), but in both cases of natural and experimental substitution reactions, the present result shows that Na was the most abundant cation. The evolution of the initial Mg-Na-Cl solution into a reductive Na-Cl brine, as proposed by Nakamura et al.[4], supports that the ionic composition of the solution in contact with phyllosilicates during the final stages of "liquid" water−rock interaction at Ryugu was also Na-rich. Although direct evidence for the chemical composition of

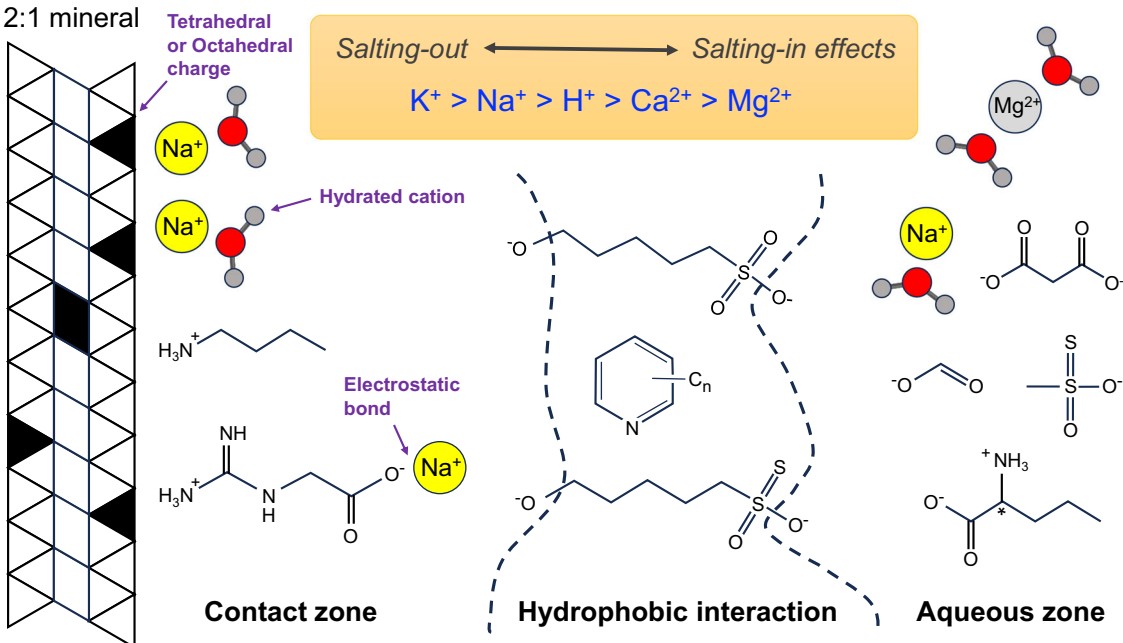

**Fig. 8 | Conceptual model to illustrate cation-organic-mineral interactions.** At the clay mineral surface contact zone, cations and amphiphilic fragments accumulate on the charged surface through electrostatic interactions, with the hydrophobic part directed toward the outer side of the polar aqueous solution. In the discontinuous hydrophobic interaction zone, the hydrophobic parts of organic molecules (e.g., alkylpyridines, shown above) and carbon chains of amphiphilic molecules (e.g., hydroxy alkylthiosulfonic acids) are shielded from the polar aqueous phase (molecular assignment by Naraoka et al.; Parker et al.; Yoshimura et al.;

Oba et al.[5,7,20,22]. In the aqueous solution zone, various organic substances and cations and anions are highly mobile. The main dissolved cation is Na, as confirmed by the ionic composition of the cation exchange pool of the clay minerals (this study), the chemical extraction of the salt fraction[7], and the chemical reaction modeling of aqueous alteration[4]. The trends of cation salting-in and -out effects, known as the Hofmeister series, are also shown. This scheme is modified after Kleber et al.[67]. It is not intended to represent all of the organic and mineral surface types involved in the actual reactions of the aqueous alteration of asteroid Ryugu.

the solution is scarce, the cation exchange pools of extraterrestrial materials are a quantitative snapshot of water chemistry. This novel archive can provide an accurate picture of the chemical interaction of the fluid and coexisting prebiotic organic molecules on Ryugu because monovalent cations tend to act as bulk electrolytes that stabilize the surface charge of molecules (Fig. 8).

The results of the sequential leaching and analysis of carbonate grains in Ryugu samples can be summarized as follows:

1. Sequential leaching revealed chemical and isotopic characteristics of each mineral with different solubility. The combination of Mg/Ca and Mg/Fe with Mg isotopic compositions ($\delta^{25}Mg$) describe the mixing and precipitation behavior of each mineral component. Generally, the more soluble mineral phases were found to be characterized as enriched in light Mg isotopes.

2. $\delta^{25}Mg$ of the breunnerite grain precipitated during aqueous alteration of the Ryugu parent body and observed mass-dependent isotope fractionation associated with carbonate precipitation succeeded in constraining the $\delta^{25}Mg$ endmember of secondary minerals more accurately than the sequential leaching of Ryugu aggregates. Evidence of mass-dependent $^{25}Mg/^{24}Mg$ fractionation of carbonates means that this indicator is a useful new tool for investigating carbonate neoformation, solute feeding mode, dissolution, and reprecipitation[e.g.48]. Carbonates and phyllosilicates are ubiquitous minerals in carbonaceous chondrites, and assuming a fractionation factor of dolomite of 0.99917, the isotope fractionation factors for phyllosilicates during the aqueous alteration of Ryugu can be constrained to 1.00027 (for a partition ratio of phyllosilicates, carbonates and exchangeable Mg of 92.8:4.9:2.3) and 1.00020 (for a ratio of 96.3:2.5:1.2). The Mg isotopic ratios of the carbonates indicate that the precipitation of phyllosilicates occurred first, followed by dolomite precipitation. Some

phyllosilicate precipitation continued, however, after the dolomite precipitation.

3. The sequential chemical leaching results revealed the composition of cations of different solubility phases and enabled the aqueous alteration history of Ryugu to be reconstructed. The ionic composition of the cation exchange pool showed that a Na-rich fluid was present at the time of the last contact with an aqueous phase. Because of the limited sample size, only tens to hundreds of nanograms of Mg were available in several of the fractions in this study, but the analytical uncertainty could be further reduced by more accurate analysis of large carbonate samples[35,38–41]. Such analysis would lead to a more accurate understanding of the end components and improved accuracy of temperature estimates.

The historic and successful sample returned to the Earth from the asteroid (101955) Bennu on 24 September 2023, the OSIRIS-REx (Origins, Spectral Interpretation, Resource Identification, Security, Regolith Explorer) sample opportunity was on the schedule[49]. The sample was safely and successfully recovered to process the laboratory-based analysis[50]. Interestingly, onsite investigations of Bennu have detected large "whitish" veins potentially associated with carbonate assemblages, which are thought to have been formed by aqueous alteration[51]. For further understanding of primordial aqueous chemistry[25,52], the chemical leaching approach of the present study (and the SOM leaching study) is expected to elucidate differences in the partitioning of primordial soluble ions during the aqueous and/or thermal history of this carbonaceous asteroid.

## Methods

### Breunnerite grain sample and laser Raman micro-spectroscopy
The largest single grain of breunnerite was physically isolated from the Ryugu C0002 sample obtained from the second touchdown location[1]

(Fig. 1). Ryugu C0002 is a large pebble sample and its elemental composition has been reported in detail[4,6,53]. We identified the carbonate grain (~300 μm) by using non-destructive laser Raman microspectroscopy[54] prior to wet-chemical treatment.

Raman measurements were performed on the C0002 carbonate grains (Fig. 2) and on carbonate reference standards by a Raman-11i microscope (Nanophoton Corp., Japan) in combination with an Eclipse Ti (Nikon Corp., Japan) microscope[54]. A Nikon Plan Fluor (40x, NA 0.75, WD 0.66) objective lens was used to focus the laser onto the sample, and the backscattered light was collected by the same objective lens. The excitation wavelength was 532 nm, and the power was 10 mW at the sample plane. Spectra were measured with 120-s exposures and averaged twice. The slit width at the entrance of the spectrometer was 50 μm, and a grating of 1200 grooves/mm was used. Under these measurement conditions, spectra from approximately −30 to +1270 $cm^{-1}$ were recorded every 1.1 $cm^{-1}$. Peak frequencies were determined to an accuracy of approximately 0.1 $cm^{-1}$ by fitting each peak with a Lorentz function after baseline correction. The repeatability of the peak frequency was generally smaller than 2 $cm^{-1}$. The baseline was defined as a linear function connecting spectral points ±50 $cm^{-1}$ from the apparent peak frequency. Two carbonate grains from C0002 were identified by comparison with the reference standard spectra of calcite, magnesite, siderite, rhodochrosite, and dolomite measured together with C0002 grains, and also by comparison with the reference peak frequencies of breunnerite, ankerite, and kutnohorite[54].

## Aggregate samples and sequential leaching

Ryugu aggregate samples were collected at the first (A0106) and second (C0107) touchdown sites on the asteroid Ryugu[5,7,20]; cf. assessing for dissolved organic matter[25]. The latter sample, C0107, potentially contained subsurface materials[55,56] from the artificially made impact crater (Fig. 1). Data on the production rates of surface-irradiated nuclides indicate that exposure conditions and depths differed between the chambers[56], suggesting that the sample from chamber A had an exposure age of approximately 4 million years, whereas the sample from chamber C was estimated to have been ejected from the lower part of a 1.7-m artificial crater[55]. The sample weights used for the sequential chemical leaching were 0.16 mg for A0106 and 0.42 mg for C0107 (Supplementary Fig. S1). Sequential leaching was performed on these samples using 500 μL of each solvent in the order $H_2O$, 1.0 M $NH_4Cl$ (adjusted to pH = 8.0), 0.25 M $Na_2EDTA$ (referred to as EDTA, pH = 6.5), 1.0 M $CH_3COOH$, and concentrated HF and $HClO_4$ in a 1:1 mixture. Samples were weighed into nitric acid-washed PFA centrifuge tubes. Then each extraction solution was added, followed by a 40 kHz ultrasonic bath for 5 min; the reaction was carried out for 2 h on a shaking table. All reactions were carried out at room temperature. In each extraction step, the supernatant was collected after centrifugation for 5 min, and the residue was dissolved in HF/$HClO_4$[57]. Note that the precipitation order during aqueous alteration is opposite to the extraction order, because precipitates are removed into the solid phase in order of decreasing solubility, whereas in each chemical leaching step, the reactivity and acidity of the reagents increases so that components with higher solubility are extracted before those with lower solubility.

SOM in the Ryugu samples was previously subjected to sequential extraction[5,7]. The samples were reacted with $N_2$ gas-purged ultrapure water (Tamapure AA100, Tama Chemical Co., Ltd. Japan) in flame-sealed ampoules that were also purged with $N_2$ gas in the headspace at 105 °C for 20 hours. Next, the sample was reacted with a mixture of dichloromethane (DCM) and methanol (MeOH) in a 1:1 volume ratio (DCM/MeOH, #8) for 15 min at room temperature, followed by >99% HCOOH (#9) overnight at room temperature. Finally, the sample was reacted with 20% HCl (#10) for 15 min at room temperature to complete the extraction of soluble material. The supernatant fluid in each

fraction was collected by centrifugation. For comparison, Tarda (C2$_{ung}$), Aguas Zarcas (CM2), and Jbilet Winselwan (CM2) (Table 1) were also subjected to similar extraction experiments by Yoshimura et al.[7] (Supplementary Table S5). Only the HCOOH and HCl extracts of these meteorites were used for analysis because the others were used preferentially for the analysis of SOM. The SOM leaching results for Ryugu and these meteorites are also presented in Table 2. In addition, $\delta^{25}Mg$ of the hot $H_2O$ extract in the SOM leaching was also measured in this study.

The present study and the SOM leaching study differed as follows. Leaching was carried out in this study under more mild conditions and with the intention to subdivide the fractions extracted by hot $H_2O$ (#7-1) and HCOOH (#9). The first $H_2O$ leaching step in the present study was carried out at room temperature, whereas the SOM leaching was conducted at high temperature under anaerobic conditions. Second, only in this study was $NH_4Cl$ used for extracting exchangeable ions, mainly from clay minerals (phyllosilicates), where adsorbed ions are recovered in the aqueous phase when they are replaced at the exchangeable sites with $NH_4^+$ ions, for which they show a high preference. The prepared $NH_4Cl$ solution was slightly alkaline, with the aim of preventing the dissolution of carbonates. In the SOM leaching, carbonates were extracted with formic acid[7], which dissolves dolomite almost completely but also dissolves a part of Mg-bearing phyllosilicates. In contrast, EDTA selectively dissolves only carbonates. Acetic acid ($CH_3COOH$), which is similar to formic acid (HCOOH), can recover residual carbonates. Note that EDTA mostly does not react with clay minerals, whereas $CH_3COOH$ may dissolve ~10% of saponite, a very soluble clay mineral (Supplementary Table S1). The SOM leaching dissolved phyllosilicates with hydrochloric acid, but silicates such as olivine were not completely dissolved. In contrast, HF/$HClO_4$ completely dissolved the residual, including silicates.

## Inductively coupled plasma mass spectrometry

Trace-element concentrations were measured by quadrupole inductively coupled plasma mass spectrometry (ICP-MS, iCAP Qc, Thermo Fisher Scientific Inc., MA, USA). A 0.3 M $HNO_3$ solution was added to each vial to dilute the samples. The $HNO_3$ used in this study was a commercially supplied high-purity reagent (TAMAPURE AA-100, Tama Chemical, Co., Ltd. Japan). We added internal standards (Be, Sc, Y, and In) to the $HNO_3$ to correct for instrumental drift. Mass interferences were eliminated by using the kinetic energy discrimination mode with $H_2$ for Fe analysis and He gas for the other elements.

For the Mg isotope analysis, each extract was dried down on a heater and then re-dissolved in 8 mM $HNO_3$. Samples were purified in an IC Metrohm 930 Compact IC Flex system coupled to an Agilent 1260 Infinity II Bio-Inert analytical-scale fraction collector system (Agilent Technologies, Santa Clara, USA) set in a class-1000 clean hood[58,59]. For complete separation of cations, the samples were eluted through a Metrohm Metrosep C6-250/4.0 column with 8 mM ultrapure $HNO_3$ at a flow rate of 0.9 mL·$min^{-1}$. Magnesium isotope ratios were measured by a multiple collector (MC) ICP-MS Neptune plus (Thermo Fisher Scientific Inc., MA, USA) at the Geological Survey of Japan, National Institute of Advanced Industrial Science and Technology. We performed Mg isotope analysis with a nickel sampler cone and a high-sensitivity X-skimmer cone. Sample solutions were introduced with a PFA nebulizer (MicroFlow, ~50 μL·$min^{-1}$, ESI, Omaha, USA) attached to a quartz dual-cyclonic spray chamber in free aspiration mode. The beam intensity for the 100 ppb (parts per billion) solutions was approximately 5.0 V for $^{24}Mg$. After initial uptake of the solutions, a single analysis consisted of 40 cycles with an integration time of 4 s per cycle. The background signal intensities were measured with a 0.3 M ultrapure $HNO_3$ solution for 1 cycle with an integration time of 30 s per cycle. The 2 SD values of each data point were calculated from triplicate measurements of each sample.

The isotopic data are expressed as per mil (‰) deviations relative to the DSM-3 standard. The Mg isotope ratio was defined as follows:

$$\delta^{25}Mg = \left[ \left( ^{25}Mg/^{24}Mg \right)_{sample} / \left( ^{25}Mg/^{24}Mg \right)_{DSM-3} - 1 \right] \times 1000 \quad (2)$$

### Carbon (C) and nitrogen (N) contents and their isotopic compositions

For assessing the carbonate-bearing potential, we compiled the elemental abundance of carbon (C, wt%) and nitrogen (N, wt%) and the $\delta^{13}C$ (‰ vs. VPDB) and $\delta^{15}N$ (‰ vs. Air) isotopic compositions (Table 1). For the total C and N contents with their isotopic compositions ($\delta^{13}C$, $\delta^{15}N$), we used an ultrasensitive nano-EA/IRMS method (Flash EA1112 elemental analyzer/Conflo III interface/Delta Plus XP isotope-ratio mass spectrometer, Thermo Finnigan Co., Bremen)e.g.[5,20,60]. In Table 1, data of the Ryugu aggregate samples (A0106, C0107) and of reference carbonaceous meteorites are presented. The C and N isotopic compositions (δ values) are denoted in relation to international isotope standards as follows:

$$\delta^{13}C = \left[ \left( ^{13}C/^{12}C \right)_{sample} / \left( ^{13}C/^{12}C \right)_{VPDB} - 1 \right] \times 1000 \quad (3)$$

using the Vienna Peedee Belemnite (VPDB) standard.

$$\delta^{15}N = \left[ \left( ^{15}N/^{14}N \right)_{sample} / \left( ^{15}N/^{14}N \right)_{Air} - 1 \right] \times 1000 \quad (4)$$

using the Earth atmospheric nitrogen (Air) standard.

## Data availability

The Hayabusa2 project is releasing raw data on the properties of the asteroid Ryugu from the Hayabusa2 Science Data Archives (DARTS, https://www.darts.isas.jaxa.jp/planet/project/hayabusa2/) for Optical Navigation Camera (ONC), Thermal InfraRed Imager (TIR), Near InfraRed Spectrometer (NIR), LIght Detection And Ranging (LIDAR), SPICE kernels, and PDS4. We declare that all these database publications are compliant with ISAS data policies (https://www.isas.jaxa.jp/en/researchers/data-policy/). The data generated in this study are provided in the Supplementary Information/Source Data file.

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

## Acknowledgements

The Hayabusa2 project was led by ISAS (Institute of Space and Astronautical Science)/JAXA (Japan Aerospace Exploration Agency) in collaboration with DLR (German Space Center) and CNES (French Space Center), and supported by NASA (U.S. National Aeronautics and Space Administration) and ASA (Australian Space Agency). We thank the members of the Astromaterials Science Research Group (ASRG) at ISAS, and the Hayabusa2 curation team for conducting the sampling and quality control management. We also thank Dr. C.Yoshikawa (JAMSTEC) for the assessment of isotope fractionation model; Y.Yoshikawa (JAMSTEC) for laboratory assistance; Mr. Kumazoe (Kyushu Univ.) for solvent extraction of the Tarda, Aguas Zarcas, and Jbilet Winselwan meteorites;

and Y.Kobayashi (Metrohm Japan) for technical support with the ion chromatography. This study was partly conducted as part of an official collaboration agreement between JAMSTEC and HORIBA Techno Service Co., Ltd. This research was partly supported by the Japan Society for the Promotion of Science (JSPS) under KAKENHI grant numbers 21H01204 (T.Yoshimura), 21J00504 (TK), 20H00202 (HN), and 21H04501&21H05414&21KK0062 (Y.T.). J.P.D. and J.C.A. are grateful to NASA for support of the Consortium for Hayabusa2 Analysis of Organic Solubles.

## Author contributions

T.Yoshimura, Y.Takano, J.P.D., and H.N. conceived the study. T.Yoshimura, H.N., T.K., and Y.Takano conducted the sequential extractions. T.Yoshimura conducted the pre-treatment and IC analysis. D.A. and T.Yoshimura conducted the ICP analysis and the evaluation of the Mg isotopic composition. T.Yoshimura, D.A., M.B., and Y.Takano designed the magnesium isotope systematics. S.Tachibana and H.Yurimoto collected the carbonate grain sample during the C0002 sampling process. S.S., M.M., M.O., and S.Tanaka conducted the sampling and assessed the carbonate grain. N.O., N.O.O., H.N., and Y.Takano conducted the elemental analysis of the reference meteorite samples. T.Yokoyama, H.Yurimoto, M.B., and S.Tachibana provided the interpretation of asteroidal chemistry. M.B. assessed the Mg isotope systematics of Ryugu and the reference samples. H.N., Y.Takano, and J.P.D. designed the implementation of the SOM scheme prior to the initial analysis (until ~31 May 2022). S.Tachibana, H.Yurimoto, T.Nakamura, T.Noguchi, R.O., H.Yabuta, and K.S. led the initial analysis processes. M.A., T.Yada, M.N., K.Y., A.N., A.M., T.O., and T.U. curated the samples. M.Y., T.S., S.Tanaka, F.T., S.N., S.W., and Y.Tsuda contributed to the sample collection from Ryugu. All authors contributed to the manuscript revision and approved the submitted version.

## Competing interests

The authors declare no competing interests.

## Additional information

¹Biogeochemistry Research Center (BGC), Japan Agency for Marine-Earth Science and Technology (JAMSTEC), Natsushima 2-15, Yokosuka, Kanagawa 237-0061, Japan. ²Geological Survey of Japan (GSJ), National Institute of Advanced Industrial Science and Technology (AIST), 1-1-1 Higashi, Tsukuba, Ibaraki 305-8567, Japan. ³Department of Earth and Planetary Sciences, Kyushu University, 744 Motooka, Nishi, Fukuoka 819-0395, Japan. ⁴Creative Research Institution (CRIS), Hokkaido University, Sapporo, Hokkaido 001-0021, Japan. ⁵HORIBA Techno Service Co., Ltd., Kisshoin, Minami-ku, Kyoto 601-8510, Japan. ⁶Department of Earth and Planetary Sciences, Tokyo Institute of Technology, Ookayama, Meguro, Tokyo 152-8551, Japan. ⁷Centre for Star and Planet Formation, Globe Institute, University of Copenhagen, Copenhagen K 1350, Denmark. ⁸Solar System Exploration Division, NASA Goddard Space Flight Center, Greenbelt, MD 20771, USA. ⁹Department of Earth Science, Tohoku University, Sendai 980-8678, Japan. ¹⁰Department of Earth and Planetary Sciences, Kyoto University, Kyoto 606-8502, Japan. ¹¹Earth and Planetary Systems Science Program, Hiroshima University, Higashi Hiroshima 739-8526, Japan. ¹²Institute of Space and Astro-nautical Science, Japan Aerospace Exploration Agency (ISAS/JAXA), Sagamihara, Kanagawa 229-8510, Japan. ¹³Kanagawa Institute of Technology, Atsugi 243-0292, Japan. ¹⁴Department of Earth and Planetary Sciences, Nagoya University, Nagoya 464-8601, Japan. ¹⁵UTokyo Organization for Planetary and Space Science (UTOPS), University of Tokyo, Bunkyo, Tokyo 113-0033, Japan. ¹⁶These authors contributed equally: Toshihiro Yoshimura, Yoshinori Takano. ✉e-mail: yoshimurat@jamstec.go.jp; takano@jamstec.go.jp

