## [Peer Review File · Nature Communications]

Breunnerite grain and magnesium isotope chemistry reveal cation partitioning during aqueous alteration of asteroid RyuguEditorial Note: Parts of this Peer Review File have been redacted as indicated to remove third-party material where no permission to publish could be obtained.

REVIEWER COMMENTS

Reviewer #1 (Remarks to the Author):

Yoshimura and coauthors present new Mg isotope data from sequential leaches of Ryugu, in order to back out the chemical composition of aqueous phases on parent body. I think that the concept is well designed and the results could be of interest to a broad geochemical community. However, the paper is, in general, difficult to follow at times and the significance of the results are not well explained. To me, there is a big problem with the use of isotopic fractionation factors to back out the composition of the fluid phase; these fractionation factors are often poorly constrained and, depending on the study, can be highly variable. This will naturally have an effect on the uncertainty with which the various fluid compositions are constrained, and should be discussed here. The discussion of the exchangeable pool is also troubling to me, as the authors focus on surface sites but ignore cations taken up into the clay interlayer to balance charge – particularly important in minerals such as montmorillonite. Finally, by the end of the manuscript I am still a little unclear about the significance of the study. I'd like to see more effort to present the broader scale implications of this research and why readers of Nature Communications should care. I understand that samples of Ryugu are scarce and very valuable and this can potentially tell us a lot of information about processes in the early Solar system. However, there still needs to be effort to show us why these analyses are important.

If the authors address this feedback and the more specific comments below, I think it would greatly improve the paper and have no problem with it being published.

Line by line comments:

L118: I don't know that these fractionation factors are well established, and are likely to be different depending on clay mineralogy or carbonate composition.

L124: expected to differ

L125: do you mean the composition of the most recent solution to be contact with the clay minerals?

L130: To date, just one study has applied the sequential solvent extraction method...

L136: ...the Mg isotopic composition of breunnerite grains precipitated during...

L230: If a bulk Ryugu sample was not analyzed in the current study, it would be useful to compare the Mg isotope ratio obtained for a CI chondrite such as Orgueil, with values determined for Orgueil by Bizzarro et al., 2023.

L236: Mineral leaching experiments show some kinetic fractionation of Mg isotopes during the leaching process (e.g. Wimpenny et al., 2010). So this assumption may not be correct.

L255: why are HCl-containing ices enriched in Mg and Fe?

L264: define the cap-delta term

L276: I'm not an expert in Mg fractionation factors in carbonates, but I seem to recall there being a range of alpha values for dolomite and other carbonates (e.g. Geske et al., 2015). I think there needs to be more discussion, either here or in the supplemental, about the range of possible fractionation factors and associated uncertainty with reconstruction of the fluid composition. That uncertainty should be accounted for in the reconstructed fluid composition.

L285: This external reproducibility should be defined for your own laboratory.

L286: But aren't there other factors that would overprint small isotopic differences controlled by temperature (e.g. mineralogy, fluid composition)

L293: Iron is not mentioned in this section, so this should be renamed

L295: Hasn't this introduction already been made earlier in the manuscript? If so, please delete this sentence.

L299: Figure 4 shows the Mg isotopic composition vs Mg/Fe. I don't see a linear correlation between $\delta^{25}\text{Mg}$ and Mg/Ca in Bizzarro's samples. I also don't understand the explanation here. The endmember dolomite composition was calculated from the correlation between $\delta^{25}\text{Mg}$ and Mg/Fe to be -1.4 to -1.33permil. And it was deduced this way because analysis of a dolomite grain was not possible. But on L302 you then state that this endmember composition is lower than the composition of dolomite. Do you mean the breunnerite grain? I'm very confused by this section. Also, Fig. 4 is difficult to understand, and the caption is way too long (and for other figures).

L305: even if any of this made sense, what is the uncertainty associated with the dolomite endmember composition? An isotopic difference of 0.3 to 0.4 permil is quite small, can you be sure that it is significant?

L314: This was not observed. It was calculated based on assumptions.

L316: So phyllosilicates precipitate first, followed by dolomite and then more phyllosilicates? What evidence is there for such a precipitation order?

L324: This is quite vague – do you mean that the carbonate leaches are isotopically heavier than expected? If so, state that here.

L336: What is the SOM leaching?

L334-352: This is a rambling discussion that doesn't really mention Mg. A lot of the information provided is not required, instead it would be better to focus on what Mg is doing and use the behaviour of other elements to support the explanation.

L364: Is that the current consensus with regards to exchangeable Mg or is there debate in the literature?

L370: what are the other solutions? The progressive leaching removes exchangeable cations before attacking carbonates and finally any silicate materials. So chemical and isotopic differences reflect that different reservoirs are being targeted, rather than changes in the solution composition. You have tried to back out the composition of the aqueous phase during carbonate precipitation but this is not what is shown in the ternary diagram or the measured element ratios.

L382: What do the arrows signify? I would have thought smaller ions (Li and Mg) are more likely to diffuse into the interlayer region of a clay mineral than large ions such as Ba.

L388: How is this selective adsorption constrained and what is the mechanism?

L391: In many clay minerals, particularly expandable clays like smectite, the majority of exchangeable ions enter the interlayer to balance charge, rather than being chemically bound. These minerals have a far greater cation exchange capacity than non-expandable clays such as kaolinite. Have you considered interlayer expansion and uptake of ions into this region? The later discussion of the dissolved composition is solely based on the surface layer exchange, which may be misleading.

L397: Don't these partition coefficients essentially mean that Na will always dominate the cation pool no matter what the composition is? What are the average element ratios in the exchange pool? It would be useful to see these for some context.

L398: ...using the average element ratios...

L400-408: Again, this discussion should be simplified; do we really need all of these ratios? The problem is that the thrust of the discussion is lost. I am not sure why any of this is important.

L412: Is this a surprise? Na has long been known to be very mobile during weathering.

L426: The breunnerite grain is barely mentioned. To be honest, I had forgotten that you had analysed it.

L430: I don't understand this – the sequential leaching is the backbone of this paper! Was it not very good? In that case, what is the point of the paper?

L435: This is the first mention of any Mg isotopic fractionation factor. This should be introduced and explained in the discussion.

L437: See earlier comment – what is the evidence for continued phyllosilicate precipitation?

L443: See earlier comment about the KD values used here – they would always select for Na preferentially in the modeled solution.

Reviewer #2 (Remarks to the Author):

See pdf file.

[Editorial Note: This file is displayed over the next nine pages]

Review of manuscript NCOMMS-23-47707, by Yoshimura et al., entitled, “Breunnerite grain and magnesium isotope chemistry within cation-partition dynamics during aqueous alteration of asteroid Ryugu”

Summary of the paper

This paper is investigating the alteration history of Ryugu as viewed by Mg isotopes. Using a step leaching sequence from most soluble to least soluble, the authors can have sequentially access to the Mg isotopic composition of the exchangeable ions, carbonates and silicates, and analysed as well a breunnerite grain. They show a progressive enrichment in ^{25}Mg from most soluble phases to least soluble phases. Based on a simple model of phyllosilicate-carbonate precipitation, they showed a 3 steps alteration with about 70% of Mg uptake from phyllosilicate precipitation followed by mixed phyllosilicate-carbonate precipitation and finally a last step of phyllosilicates. Using the most soluble leachate, they estimate the composition of the final fluid composition, and better understand the behaviour of the different phases in solution (organic matter for instance).

Overall Impression

This paper is rather well written and propose an interesting approach for deciphering fluid alteration characteristics and conditions. Despites being most of the time well written, some sentences/paragraphs could be rephrased (and some sentences cut in half) to facilitate the reading. The current version is every now and then a bit hard to follow as figure references are not always correct, but the results and discussions are well illustrated by numerous figures. The discussion is highly structured which help to a good understanding. However, some more details can and should be provided in order to be able to reproduce model presented in this article and have a broader discussion. I think it would beneficiate from a bit of careful work on clarifying points and correcting some mistakes. That being said, the content of the paper is interesting, and the community will beneficiate for this high-quality study. Therefore, I recommend publication in Nature communications after major revisions.

General comments

Model of phyllosilicate and carbonate precipitation should be better details to be reproducible.

Further discussion about the interpretation of Mg-isotopic composition of leachate should be done (some details are already given in caption of Fig. 4).

You model only a narrow range for carbonate precipitation. You should explain in more details why? Are you expecting only a short range for dolomite/carbonate precipitation or could you envision a wider range but we only access to average due to the nature of analysis?

The data could be better synthesised at least in supplementary table and clearly identified with new data from this paper along with literature data necessary to the conclusions of this paper, also in figures (e.g., Fig. 2). Besides, I do not see real contribution of Table 1 in this study.

Reference to table or figure is sometimes wrong so that it can be complicated to really follow what the authors want to tell us. Please have a careful look to the reference to figure, figure panel and table.

Some references are cited as 2023 but are 2022. Please correct accordingly throughout the text. Some other references might also be 2022 and not 2023, please have a careful look on the references.

It is ok for the review, but lots of “—” sign are not on the same line as the figures. Please be careful during the proof stage to that point.

Detailed comments (line numbers refer to the beginning of a sentence)

Main Text

Intro

L75: “radiative”, I think it is rather “radioactive”.

L81: Nakamura et al., 2023 might by 2022. Please check reference.

L91: “see Table 1 for carbon abundances and isotopic compositions”. I do not think this is the appropriate place for this reference. You speak about mineralogy. Table 1 is volatile content and isotopic composition. Also, clarify the input of data in table 1 for this study. Such data are not really discussed in the text.

L94: “breunnerite [(Mg, Fe, Mn) (CO₃)₂]”. I think there is only 1 CO₃. Please correct accordingly.

L120: “Given that the isotope fractionation factors (α) for carbonate ($\alpha < 1$) and clay ($\alpha > 1$) minerals are generally opposite in sign (Saenger and Wang, 2014; Wimpenny et al., 2010, 2014), the $\delta^{25}\text{Mg}$ composition of the residual liquid phase is expected **to** differ from that of the starting solution according to which mineral precipitates first.”

- Please at “to” between expected and differ (bold).
- I am not sure that I understand the implication between the two part of the sentence. In any case, if there is isotope fractionation of a single species, the residual liquid phase is expected to differ from that of the starting solution. Do you want to say that as there are two species involved with opposite fractionation factor, the evolution of the residual liquid phase will depend on which phase is precipitating first? I would also mention the evolution is function of the values of the fractionation factors and the proportion of each phases to precipitate.

Please clarify this sentence.

Results

L149: it if Fig 2C. I will try to tell you the other one, but please check all your figure 2 references since you seemed to have added a panel to that figure without updated the references.

L150: You describe breunnerite as [(Mg, Fe, Mn) (CO₃)], saying Mg, Fe and Mn are the main compounds of the crystal, but Ca is actually more abundant than Mn and not discussed (Fe/Ca = 2.45; Ca/Mn = 3.10). Why Ca is not in the formula? Could this grain actually be a small-scale mixture with calcite?

L150: Is breunnerite a super group that encompasses calcite, siderite, dolomite and rhodochrosite? It is not really clear when you say breunnerite because there is no breunnerite standard in the supplementary figure S2.

L150: Can you provide the XRF spectra, at least in the supplementary material, as well as elemental compositions for the breunnerite.

L153: "(Figs. 2C and 2DP)." Please delete the "P". Update reference: 2D and 2E.

L154: Can you give extend of peak position? It is hard to really read that information from the figure.

What is the precision (reproducibility) you have on the position of the peak?

L155: Update reference: 2D.

L156: Update reference: 2E.

L157: It seems that the variation of the peak position for the breunnerite peak and the carbonate peak are independent. Can the surface irregularities affect the position of both peaks differently?

L158: Please add "quadrupole" before inductively coupled plasma mass spectrometry (ICP-MS) as in the material and method part.

L160: I am not sure "whereas" is needed here.

L164: For non-aware reader, this paragraph could be hard to follow because they do not know what is extracted by each solvent. It could be a good idea to summarize what is expected (maybe with the help of Supplementary Table S2).

L178: "see Supplementary Fig. S3 for the $\delta^{26}\text{Mg}^*$ value in each leachate in this study and in the SOM leachates reported by Yoshimura et al., 2023)." I would rather do a sentence out of the brackets saying that you observe not ^{26}Mg -excess in the leachate. E.g., " $\delta^{26}\text{Mg}^*$ value in each leachate in this study and in the SOM leachates reported by Yoshimura et al., 2023 are indistinguishable from 0 ‰ within errors (Supplementary Fig. S3)". Or "No ^{26}Mg -excess is observed in each leachate in this study and in the SOM leachates reported by Yoshimura et al., 2023 (Supplementary Fig. S3)". Actually you have a good explanation in supplementary material but I would nonetheless do a one sentence summary of that in the main text.

L198: It is not clear which are "these extracts". I would put back the names of extracts you mention.

L201: "high Mg/Ca". Is it high in general or high in comparison with the other? Because it seems to be pretty similar compared to H_2O and NH_4Cl . Please correct accordingly. Mg/Fe is intermediate. Please give more details about what you mean.

Discussion

²⁵Mg/²⁴Mg homogeneity of bulk carbonaceous chondrites and mass-dependent Mg isotope fractionation in carbonates

L220: Yes, indeed $\delta^{26}\text{Mg}$ is affected by radioactive decay of ^{26}Al . However, the deviation should be on the order of 10s of ppm (Al/Mg ratios close to solar ratio or below), so well below the mass dependant variation you should observe on the permil/0.1 permil level.

“However, ^{26}Mg is affected by the radioactive decay of ^{26}Al . Therefore, $\delta^{25}\text{Mg}$, which” I would rather note $\delta^{26}\text{Mg}$ to be consistent with the next sentence.

L242: Do you mean supplementary figure S1? Please correct accordingly.

L245: “which is as low as that of terrestrial carbonate precipitates”. Do you mean the deviation compare to the bulk is similar to terrestrial carbonate or the absolute value? If it is absolute value, what does this mean? Please clarify in the text.

L245: I am not sure about the meaning of the evolution trend. I agree that is an observation, but for further interpretation you would need to take into account fractionation factors, as you do in following part of the discussion. You can add a sentence to invite the reader to be cautious about direct interpretation.

L247: Please also give error for the Mg isotopic composition. Please also clarify the number of grains measured and if the reported value is the average of multiple measurements.

Insights from $\Delta^{25}\text{Mg}$ profiles and in situ temperature

L264: It is the first time you use $\Delta^{25}\text{Mg}$, please define it.

L267: chondrite, should be plural.

L281: “in the case of inorganic carbonates precipitated from solutions”, precipitated might be precipitation. Please read carefully this sentence and correct accordingly.

L282: Please define “saturation indices Ω ” for non-specialised people.

L291: Do you think it would be possible to estimate temperature variation from $\delta^{25}\text{Mg}$ of breunnerite. I mean T dependency is low so you would need a good precision as well as a good estimate of the $\delta^{25}\text{Mg}$ of the fluid from which the breunnerite precipitate from.

Magnesium and iron profiles of carbonates: precipitation order of Mg-bearing secondary minerals

L299: Why you are not showing a $\delta^{25}\text{Mg}$ and Mg/Ca plot? It seems to be an important plot to understand the data and your discussion. It might be a 2-panel figures along this your current Fig. 4.

L301: in supplementary table S3, please explicit what is “-“. No data available/calculated?

L302: During the first reading, it is not clear how you calculate the theoretical dolomite data.

You can add in bracket $\delta^{25}\text{Mg}_{\text{Dolomite}} = \Delta^{25}\text{Mg}_{\text{dolomite-aq}} + \delta^{25}\text{Mg}_{\text{Fluid}}$. But it should also be clearer if you define $\Delta^{25}\text{Mg}$ when using it the first time as said in a previous comment.

Actually, some details are given in the figure 4 caption. It should be also mentioned clearly in the main text. Besides "Theoretical dolomite" in Fig. 4 is not explicitly denoting that it is theoretical in a way of early precipitation if nothing else occurs before that.

You will gain to clarify this whole paragraph by better explicating the situation you consider.

L304: Please give here $\alpha/\Delta^{25}\text{Mg}_{\text{Phyllosilicates-aq}}$ for phyllosilicates ($\Delta^{25}\text{Mg}_{\text{Phyllosilicates-aq}} = +0.28 \text{ ‰}$ from Fig. 6).

L309: Can you precise what do you mean by Mg partitioning ratio of 9:1. Is it the total partitioning of Mg between phyllosilicates and dolomite, or during precipitation of dolomite?

L310: During the first read, it was not clear what was the cause and consequence. Maybe you want to say you need -0.38 ‰ decrease of $\delta^{25}\text{Mg}$, which using a simple model leads to 70 % of phyllosilicates before carbonate precipitation.

Does the model predict 70 % of phyllosilicates before carbonates or do you need 70 % of phyllosilicate to be product so that it matched the isotopic compositions? It is not clear what is assumed and what is deduce. Please clarify.

L311: Besides, it is more about 75 % than 70 (74.3 % from my calculation). Please check also the Fig. 6.

L317: The offset of -0.55 ‰ from the first extraction to the bulk lead to only 90% of Mg 'loss'. What happen to the remaining 10 % of Mg?

Besides, I think you need to compare the Mg isotopic composition of the fluid, not of the extraction, so you would need to correct for the $\Delta^{25}\text{Mg}_{\text{phyllo-aq}}$.

Trying to reproduce the same model, accounting for $\Delta^{25}\text{Mg}_{\text{phyllo-aq}}$, this leads to an offset of -0.83 ‰ which would correspond to about 98.2/95.6 % of Mg uptake, much better than the current 90 % (depending on if I let dolomite precipitation until the end or not).

Please correct and clarify this point.

Besides, you only tell L362 that the uptake of exchangeable would occur without fractionation. You should tell that here. If true, then I come back to my first sentence, what about the last 10 % of Mg? Do you have to change the fractionation factor for phyllosilicate, which would thus delay the precipitation of carbonate to a further extend of Mg uptake before?

I do not think there are enough details about the model. Please give more information about how you define carbonate precipitation (end of the precipitation?), what correspond exactly the partitioning of Mg phyllosilicate:carbonate = 9:1 (during carbonate precipitation or in total).

L322: Please cite Fig. 4 here in this paragraph. Are all points from leachates consistent with a carbonate endmember with a $\delta^{25}\text{Mg}$ of about -1.4 ‰ ?

Is the breunnerite point consistent with such a model?

L328: Do you have any idea about the cause of the higher reactivity of phyllosilicates in Ryugu compared to terrestrial one?

L330: You can discuss more the result from the leachates. Details are given in caption of Fig. 4 but not really discussed more in the main text.

For C0107 CH₃COOH: how you can distinguish between partial dissolution of labile phyllosilicates (fractionation of Mg/Fe) vs. mixing line with a more ²⁴Mg-rich carbonate?

What would be the evolution during phyllosilicate precipitation?

What about EDTA points higher than the dolomite endmembers (for both A0106 and C0107)?

It is also only possible to explain these points by Mg/Fe fractionation or should these require carbonate endmember with less ²⁴Mg-enrichments, i.e., formed earlier in alteration history?

Partitioning of Mg²⁺ in cation exchange pools and solute compositions

L362: “From the cation exchange pool of phyllosilicates (e.g., surface sites and exchangeable interlayer sites), isotopic tracers have shown that dissolved Mg is taken up without fractionation (Wimpenny et al., 2014)” Should be said earlier? (Cf comment **L317**).

L369: “which has higher δ²⁵Mg than the olivine average (Deng et al., 2021)”. I would be cautious in that comparison as Deng et al. measured olivine from chondrules. I think it is not clear whether olivine from CIs and Ryugu are all derived from chondrules material or not. In the latter case, we do not really know the isotopic composition of such olivine.

Conclusion

L435: Fractionation factors are inverted compared to their respective phase.

Samples and Methods

Brunnerite grain sample and laser Raman micro-spectroscopy

L471: You speak about the largest single grain of brunnerite, in singular form whereas you show 2 brunnerite in Fig. 2. What about the second one? Should be plural in the text?

L480: What would be the reproducibility of the peak position?

L485: These standards (brunnerite, ankerite, and kutnohorite) were measured or only compared from other studies? If you have measured them, please add them in Supplementary Figure S2.

Aggregate samples and sequential leaching

L498: Does the quantity of solvent have an effect on leaching? You have used the same amount of solvent when the amount of sample is multiplied by about 2.5.

L493: You give the exposure age of sample from chamber A. Are you expecting cosmogenic effects?

Inductively coupled plasma mass spectrometry

L560: Can you precise in which lab the Mg isotope analyses have been performed?

L565: Do the data reported be single analysis or did you do multiple replicates?

Availability of data and materials

Does the Hayabusa2 Science Data Archives will contain your data? Otherwise, will your data be available in a repository or only as supplementary material from this paper?

References

Please note that the three following papers cited from the main text and the last one from the supplementary material are cited as submitted. Please add the correct references when they will be accepted and published.

- Bizzarro, M. et al. 2023. The magnesium isotope composition of samples returned from asteroid Ryugu. *Astrophysical Journal Letters*. (submitted)
- Naraoka, H. et al. 2023b. Hydrogen isotope compositions of the Ryugu sample and carbonaceous chondrites: Implications for origins of hydrous asteroids. *Meteoritics & Planetary Science*. (submitted)
- Takano, Y. et al. 2023. Primordial aqueous alterations recorded in water-soluble organic molecules from the carbonaceous asteroid (162173) Ryugu. (submitted)
- Yada, T. et al. A curation for uncontaminated Hayabusa2-returned samples in Extraterrestrial Curation Center of JAXA: From the beginning to nowadays. *Earth Planets Space* (submitted).

These papers are 2022, not 2023. Please check carefully the references for possible other year mismatched.

- “Yokoyama, T. et al. 2023. Samples returned from the asteroid Ryugu are similar to Ivuna-type carbonaceous meteorites. *Science* 379, eabn7850.”
- “Nakamura, T. et al. 2023. Formation and evolution of carbonaceous asteroid Ryugu: Direct evidence from returned samples. *Science*, eabn8671.”

Supplementary material

List of Abbreviations: I would do a list rather than a paragraph for clarity.

Supplementary text

L140: Fractionation factors for phyllosilicates are not consistent between main text and supplementary material: 1.00028 in main text and 1.00054 in supplementary material. Which one have you used? Please correct accordingly. Besides, if the first one is correct, it is then quite different compared to Wimpenny et al., 2014. Can you comment?

Besides, give more details about how you estimate these two values.

L172: “ β value of 0.511”. This beta is kinetic. 0.521 is for equilibrium (Davis et al., 2015). Please correct the text with the good value. Have you used 0.521 for calculating the $\delta^{26}\text{Mg}^*$?

L198: “adsorptive”, is that correct?

Supplementary table S1: You can delete the 2nd header of the table. There is no need. Or you need to change “Na+K” to only “K” for the 2nd header?

I would add ratios discussed in “*Ionic composition of the fluid in contact with cation exchange pools*” part as well as average and reconstructed fluid compositions.

I think this table is pretty important and could be also insert in the main text.

Supplementary Figure S2: You misspell the name of the orange spectrum rhodochrosite. It misses an “h”. Besides, why you do not observe the features above 1200 cm^{-1} compared to the Caltech reference?

Supplementary Figure S3: Can you add $\delta^{26}\text{Mg}^*$ for Ruygu from Bizzarro et al., 2023? Can you also add the breunnerite sample?

Tables

Table 1: Data in this table are not really discussed in the text. What is the interest of C, N and H content on bulk samples for the step leaching?

Table 2: Can you calculate the $\text{Mg}/(\text{Mg}+\text{Fe})$ for breunnerite. I would add in the title of the table “and breunnerite”, as it is not really an extraction. Have you measured the Mg/Al ratio for the breunnerite? It would be also interesting to plot it in the Supplementary Figure S3, if measured.

Please add Orgueil data from Fig. 3 (cf comment of Fig. 3).

In caption of Fig. 2, it is written “Then, a high-precision analysis of the magnesium (^{24}Mg , ^{25}Mg , ^{26}Mg) isotope systematics of the two grains was carried out.” It is not clear if you have measured separately the two grains or not. Please add details in the text and put the 2 values in Table 2 if you measured them separately.

I would add in this table the content of the Supplementary Table S1, as it is the subject of one part of the discussion.

Figures

Fig. 2: Can you remove the thin grey contour in panel D and E?

Fig. 3 (L674): “The yellow circle with the cross represent the data from the #5 H₂O extract of Cl-group Orgueil; these data are consistent with the #7 hot H₂O data (Supplementary Figure S1).” I do not understand which is which. Why the H₂O extract is in the hot H₂O extract? Please also give the value for Orgueil as a comparison in Table 2.

Why not add the breunnerite(s) data in the Fig. 3? I would be nice to have a figure with all data, and you can easily extant the range of Fig. 3.

There are also blue-green points for CMs and CVs. Please correct the caption. Why a special highlight to the Bizarro et al., 2023 data?

You can add an arrow to indicate increase in solubility/order of precipitation to help the reader, e.g., to support discussion L245.

Please indicate with more detail what is extracted for carbonate. You give such detail in supplementary figure S1.

Please clearly differentiate your data from previous data (e.g., using "*" for previous data).

Fig. 4. Why not show the H₂O and NH₄Cl step also?

"Mixing curves for the residue and the carbonate end components are shown in gray and red, respectively." Not sure to understand what you mean. The gray mixing curve is also heading toward carbonate composition but of breunnerite.

You should add a Mg/Ca plot as it is also discussed in the text. This would be a great support for the discussion.

Besides, for simplifying the legend, I would use a symbol legend with only solvent name and a colour legend with Ryugu grains name.

Please precise to what correspond the number on the mixing line: fraction of Mg coming from carbonate or fraction of carbonate compared to phyllosilicate (that is to say taking already into account the 9:1 ratio between phyllosilicates and carbonates)?

L708: Were breunnerite would fit in the order of fractionation factors? What is the order of variations between the fractionation factor for the cited minerals?

Fig. 6. As also mentioned in the text, please better explain the model and how you define carbonate precipitation, here, in the main text or in the supplementary material.

Please verify the extend of each region (cf comment **L311**).

Reviewer #3 (Remarks to the Author):

Review Nature Communications, NCOMMS-23-47707

Yoshimura et al.

This manuscript presents Mg isotopic compositions of leachates from asteroid Ryugu to investigate its aqueous alteration history. The authors observe a significant variation of the Mg isotopic compositions between the different leachates. The results show an enrichment in the lighter Mg isotopes in the breunnerite grains, the leachates with weak acid show less enrichment and the leachates with the strongest acid are the most depleted in lighter isotopes of Mg and corresponds to the bulk composition. From this data and input from the literature, the authors concluded that 70% of the Mg was precipitated into the phyllosilicates followed by dolomite precipitation and that the last solution that was involved in the alteration of Ryugu was Na-rich.

While the introduction is well written and the objectives of the study are clear, I believe that the discussion part is hard to read and not well organised, this part would require some clarification. Furthermore, important data considered in this manuscript and, used in the interpretation of the results, are from Bizzarro et al., 2023 which is only a submitted work. Until this work is accepted, I am afraid that I can not advise on the publication of this paper. Two others cited papers are also only submitted work (Takano et al., 2023 and Naraoka et al., 2023) but their data are not used in this manuscript. Overall, the interpretation of the data seems accurate (though it is not my speciality) and worthy to be published to Nature Communications.

Hence, I approve the submission of this paper if the discussion is re-written to make it more clear and straightforward and after the manuscript of Bizzarro et al., 2023 is accepted.

Minor comments

Line 57: Why using two different notation ($^{25}\text{Mg}/^{24}\text{Mg}$ and $\delta^{25}\text{Mg}$) to present Mg isotopic composition?

Line 123: “the ^{25}Mg composition of the residual liquid phase is expected differ from that” do the author mean “is expected to differ”?

Line 164: “the $\delta^{25}\text{Mg}$ was highest” is a bit clumsy. I would say: The most enriched sample in heavy Mg isotopes is etc. (same comment for lines below)

Line 178: “ $\delta^{26}\text{Mg}^*$ ” What does the * represents here? If radiogenic why present this without talking about it?

Line 286: I believe that in order for Mg isotopes to be an indicator of temperature, one need to know if the isotopic fractionation is under equilibrium or kinetic conditions, which is not the case in this study.

Replies to Reviewer's comments on the manuscript

NCOMMS-23-47707

We appreciate the constructive comments on our manuscript (# NCOMMS-23-47707) entitled “**Breunnerite grain and magnesium isotope chemistry within cation-partition dynamics during aqueous alteration of asteroid Ryugu**” (by Yoshimura et al.). We carefully read all of the comments in the reviews and modified the manuscript based on your helpful feedback. The changes that we made based on the comments are shown in red text in the revised manuscript/supporting information. We have chosen the option of "Transparent peer review", to make public the discussion during the review process. We believe our point-by-point responses to each comment will clarify the entire context of the paper.

Reviewer #1 (Remarks to the Author):

Yoshimura and coauthors present new Mg isotope data from sequential leaches of Ryugu, in order to back out the chemical composition of aqueous phases on parent body. I think that the concept is well designed and the results could be of interest to a broad geochemical community. However, the paper is, in general, difficult to follow at times and the significance of the results are not well explained. To me, there is a big problem with the use of isotopic fractionation factors to back out the composition of the fluid phase; these fractionation factors are often poorly constrained and, depending on the study, can be highly variable. This will naturally have an effect on the uncertainty with which the various fluid compositions are constrained, and should be discussed here. The discussion of the exchangeable pool is also troubling to me, as the authors focus on surface sites but ignore cations taken up into the clay interlayer to balance charge – particularly important in minerals such as montmorillonite. Finally, by the end of the manuscript I am still a little unclear about the significance of the study. I'd like to see more effort to present the broader scale implications of this research and why readers of Nature Communications should care. I understand that samples of Ryugu are scarce and very valuable and this can potentially tell us a lot of information about processes in the early Solar system. However, there still needs to be effort to show us why these analyses are important.

➔ The influence of interlayer ions in phyllosilicates, especially in saponites belonging to the high CEC montmorillonite is of great importance and the following explanation has been

added to the current version. Isotopic fractionation of exchangeable ions in clay minerals is discussed in detail in Wimpenny et al. (2014, *Geochim. Cosmochim. Acta*, vol. 128). They reported no significant fractionation between ambient dissolved Mg^{2+} and Mg incorporated into the mineral's exchangeable pool in any of the layered silicates of montmorillonite, illite and kaolinite. This exchangeable pool, which may have been misleading with regard to the description of surface sites, includes all Mg pools adsorbed at interlayer and surface sites by electrostatic interactions, and refers to components that can be extracted by NH_4Cl and other means. In the data we have presented here, we have also tried to extract the replaceable pool. However, we did not indicate whether there is really no influence of structurally substituted components that cause isotopic fractionation, so we have added an explanation. As for the substitutable Mg present in the layered silicate sheet, it is isotopically fractionated, as you have pointed out. With regard to this quantitative contribution and its effect on the isotopic composition of the exchangeable fraction, we have added the following discussion to the Supplementary Information:

“Neutral solutions of NH_4Cl should not cause breakdown of the clay structure (Wimpenny et al., 2014), but the Ryugu sample is highly reactive and the mineral lattice may be slightly broken by reaction with the solution. The exchangeable pools of phyllosilicate minerals consist of Mg adsorbed on interlayer and surface sites by electrostatic interactions. These adsorbed cations are relatively weakly bound and can be easily exchanged by using NH_4Cl . In contrast, in the smectite family, which includes saponite, a typical Ryugu secondary silicate mineral, the majority of Mg is structurally bound due to isomorphous exchange of Mg^{2+} and Al^{3+} in the octahedral layer. Therefore, as the structural Mg contamination could be due to the dissolution of the phyllosilicates themselves, we tried to estimate the influence of the structural component of the saponite from the Al concentration in our extraction solutions.

The Mg/Al ratios of the solutions are shown in Table 2. The highest Mg to Al volume ratio is found in the NH_4Cl extract, where Mg is 36 times higher in A106 and about 650 times higher in C107. This may reflect the fact that the NH_4Cl solution successfully extracts the exchangeable ions without causing significant dissolution of the clay. Even assuming for simplicity that the structural Mg substituted with Al in a 3:2 ratio to keep the layer charge neutral is dissolved by congruent dissolution, the associated release of structural Mg is a few percent of the total Mg extracted. Furthermore, other cations are also present at the substitution site. The cation structurally replacing the Al^{3+} is most likely not only Mg, but also other cations such as Na. The main cation composition of the solution during actual aqueous alteration is shown in Supplementary Table S1, where Mg only accounts for 12.6%

to 21.9% of the exchangeable fraction. Thus, the substitutable Mg would only have a low contribution to the exchangeable pool of Mg. Indeed, the Mg isotope ratios ($\delta^{25}\text{Mg}$) of these samples agree well at $-0.67 \pm 0.09\text{‰}$ and $-0.61 \pm 0.04\text{‰}$, respectively. The good agreement of the isotope ratios despite the large Mg/Al difference between the two samples supports the small influence of structural Mg.

On the other hand, the first extract, which represents the ultrapure water fraction, gives lower Mg/Al than the NH_4Cl extraction, namely 4.58 times Mg to Al in A106 and 16.52 times Mg to Al in C0107. The effect of silicate dissolution is about an order of magnitude greater in these samples than in the NH_4Cl fraction. The $\delta^{25}\text{Mg}$ of C107 is $-0.68 \pm 0.09\text{‰}$, in good agreement with the NH_4Cl extract, while A106 has a slightly lower value of $-0.85 \pm 0.02\text{‰}$. The possibility of a larger contribution of silicate to A106 can be ruled out, since $\delta^{25}\text{Mg}$ should be higher if secondary silicate dissolution is affected. In contrast, dolomite has a low value of -1.35‰ (Bizzarro et al., 2023; This study), which suggests that it may have dissolved slightly. In this case, the Mg/Ca ratio should approach 0.948 mol/mol, but the Mg/Ca of the ultrapure water fraction is 9.66 and no such trend is observed. This slightly lower $\delta^{25}\text{Mg}$ may therefore be due to slight isotopic heterogeneities in the clay mineral structure itself, with surface-bound Mg reported to have a predominantly lower isotopic ratio than the bulk and to be preferentially released by weak acids (Wimpenny et al., 2014). Moreover, extraction with hot H_2O also selectively releases components with similar isotopic compositions, which may contribute to highly soluble salt materials (Yoshimura et al., 2023). Thus, there may be isotopic differences between surface-bound and interlayer cations, but in any case the effect of secondary silicate heterogeneity on the -0.71‰ isotopic composition of the exchangeable pool we determined (Supplementary Table S4) is minor, and the effect on the Mg distribution shown in Fig. 7 is limited compared to the carbonate fractionation factors discussed below.”

The isotope fractionation factors for the clay minerals themselves are not quoted values, but calculated values obtained by assuming closed-system isotopic fractionation, which also agree well with experimental and natural weathering products, and with observed fractionation factors in groundwater. Please refer to the response on this given to reviewer 2.

L118: I don't know that these fractionation factors are well established, and are likely to be different depending on clay mineralogy or carbonate composition.

→ As you point out, the wording was changed for the composition of clay minerals and carbonates, as these affect the isotopic fractionation factors. In the present study, we do not

use specific literature values for clay minerals in the model calculation, but present our calculated values, taking into account the isotopic nature of dolomite and dissolved Mg (Fig. 6). The isotopic fractionation values for dolomite are quoted from Li et al. and were carefully constructed in the laboratory using $^{87}\text{Sr}/^{86}\text{Sr}$ and isotopic spikes. However, as pointed out by reviewer 2, we agree with the point that uncertainties in isotopic fractionation should be taken into account rather than values from a single study, and fractionation factors for natural dolomite calculated by Fantle and Higgins (2014) and Higgins and Schrag (2015), among others, are also presented. The results have been added as a supplementary figure.

L124: expected to differ

→ We have changed accordingly.

L125: do you mean the composition of the most recent solution to be contact with the clay minerals?

→ We have changed the sentence to say that the isotopic composition of exchangeable Mg is likely to reflect the fluid phase in which the clay was last reacted.

L130: To date, just one study has applied the sequential solvent extraction method...

→ We have changed accordingly.

L136: ...the Mg isotopic composition of breunnerite grains precipitated during...

→ We have changed accordingly.

L230: If a bulk Ryugu sample was not analyzed in the current study, it would be useful to compare the Mg isotope ratio obtained for a CI chondrite such as Orgueil, with values determined for Orgueil by Bizzarro et al., 2023.

→ We have added $\delta^{25}\text{Mg}$ values for Orgueil and other CIs reported by Bizzarro et al. (2023, *Astrophys. J. Lett.*) in the text for comparison with Ryugu's values.

L236: Mineral leaching experiments show some kinetic fractionation of Mg isotopes during the leaching process (e.g. Wimpenny et al., 2010). So this assumption may not be correct.

→ The primary lithology of Ryugu is composed of minerals formed by aqueous alteration. Rather than light isotopic enrichment by preferential/partial dissolution reactions determining isotopic composition, the primary minerals are thought to have reacted over a wide range.

L255: why are HCl-containing ices enriched in Mg and Fe?

→ We have corrected as follows:

“Accretion of HCl-containing ice and subsequent dissolution of primary minerals results in an initial fluid composition rich in Mg and Fe, followed by neutralization of the solution, and serpentinite, saponite and carbonate precipitation, thereby transforming it into a Na-rich alkaline fluid (Zolotov, 2012).”

L264: define the cap-delta term

→ We have added the definition of the cap-delta.

L276: I'm not an expert in Mg fractionation factors in carbonates, but I seem to recall there being a range of alpha values for dolomite and other carbonates (e.g. Geske et al., 2015). I think there needs to be more discussion, either here or in the supplemental, about the range of possible fractionation factors and associated uncertainty with reconstruction of the fluid composition. That uncertainty should be accounted for in the reconstructed fluid composition.

→ Among the carbonate minerals, dolomite is important due to its Mg content and high abundance in the Ryugu, so I would like to focus the discussion on this carbonate mineral.

Geske et al. (2015), which the reviewer pointed out, reported an apparent fractionation of Mg isotopes between modern Sabkha dolomite and evaporite pore water, with $\Delta^{26}\text{Mg}$ ranging from -0.7‰ to +0.1‰, which is considerably smaller than existing studies. They themselves concluded that this large difference in fractionation factors is related to the heterogeneous sedimentation conditions specific to Sabkha (e.g., temperature, sedimentation rate, water Mg concentration, pH) and the complex dynamics of precursor formation and dissolution/precipitation reactions induced by parthenogenetic activity (Geskes et al., 2015). Using an intermediate value for the fractionation factor for dolomite obtained at Sabkha, the fractionation factor for secondary silicates to achieve this would be 1.00040. This is consistent with the 1.00003-1.00020 (Teng et al., 2010) and 1.00029 (recalculated value of Huang et al. 2012 data by Wimpenny et al. 2014) reported for natural clay formation processes, the fractionation factor for deep groundwater and clay formation of 1.00015 (Zhang et al, 2018), and an experimentally determined fractionation factor of 1.00026 (Wimpenny et al., 2014) for the layered mineral bluecite, which is significantly higher than. Furthermore, it is difficult to reproduce the $\delta^{25}\text{Mg}$ of dolomite reported by Bizzarro et al. (2023, Astrophysical Journal Letters). For this reason, we considered that the fractionation factor for dolomite in Sabkha is probably not a sufficiently equilibrated value and did not use it in our calculations.

The isotopic fractionation of dolomite in sediments has been calibrated based on

natural water-sediment reactions by Higgins and Schrag (2015), Fantle and Higgins (2014), and others, providing insight into the isotopic fractionation of highly stabilized dolomite. The experimental fractionation factor obtained by Li et al. (2015) was applied to the calculations presented in our Figure 6. However, as mentioned above, different conditions should be considered, so we applied the fractionation factor for sedimentary dolomite, α value for $^{26}\text{Mg}/^{24}\text{Mg}$ is 0.99980 (Fantle and Higgins, 2014; Higgins and Schrag, 2015), and calculated the isotopic fractionation results. A new supplementary figure has been added to the text. An isotopic fractionation factor of 1.00029 was obtained for secondary silicates, which is in good agreement with the fractionation factors for clay minerals described above. The use of the sedimentary dolomite values suggests a slightly earlier time of dolomite precipitation. Oxygen isotope ratios of the Ryugu dolomite indicate that it was formed at high O fugacities during retrograde cooling (Fujiya et al., 2023, Hayabusa 2023 Abstract) in the late stages of aqueous alteration (Fujiya et al. 2023, Nat. Geosci., vol. 16), and our results that Mg was precipitated late in the process of partitioning into secondary minerals may be valid for both results. However, a difficulty with calibration using pore water and sediments is that there are potential contamination and advection effects, and although fractionation factors are calculated by modeling fluid movement, the influence of the magnitude of apparent isotopic fractionation cannot be ruled out. For this reason, we would like to state in the text that the figures presented in the text show the results using the fractionation factor of Li et al. (2015), as before, and the supplements show the results of other studies.

We have also added calculations for the Mg removal rate associated with precipitation, set at 1.0 and 0.5 times the rate for clay minerals (Supplementary Figs. S6C and S6D). A practical consequence of varying the sedimentation rate is the potential for greater microscale isotopic heterogeneity. Because the $\delta^{13}\text{C}$ of Ryugu dolomite is very homogeneous (Fujiya et al., 2023, Nat. Geosci.), setting a very long precipitation period may not be appropriate. Conversely, a very short precipitation period of the dolomite may also cause a spike in $\delta^{25}\text{Mg}$ of dissolved Mg due to the Rayleigh effect. These pros and cons can be evaluated in the future from the heterogeneity of Mg isotope ratios between Ryugu dolomite particles.

The above summary has also been added to the Supplemental Information.

L285: This external reproducibility should be defined for your own laboratory.

→ We have changed this sentence accordingly.

L286: But aren't there other factors that would overprint small isotopic differences controlled by temperature (e.g. mineralogy, fluid composition)

→ In relation to what you pointed out in line 274, I added that the question of which report

presents the fractional factor is more influential as a practical matter at the time, and that we need to be careful in this respect.

L293: Iron is not mentioned in this section, so this should be renamed

→ We agree with the comments and removed the part related to iron as follows:

Precipitation order of Mg-bearing secondary minerals

L295: Hasn't this introduction already been made earlier in the manuscript? If so, please delete this sentence.

→ We have removed duplicate sections from the introduction section.

L299: Figure 4 shows the Mg isotopic composition vs Mg/Fe. I don't see a linear correlation between $\delta^{25}\text{Mg}$ and Mg/Ca in Bizzarro's samples. I also don't understand the explanation here. The endmember dolomite composition was calculated from the correlation between $\delta^{25}\text{Mg}$ and Mg/Fe to be -1.4 to -1.33permil. And it was deduced this way because analysis of a dolomite grain was not possible. But on L302 you then state that this endmember composition is lower than the composition of dolomite. Do you mean the breunnerite grain? I'm very confused by this section. Also, Fig. 4 is difficult to understand, and the caption is way too long (and for other figures).

→ We have caused some confusion. The linear relationship between isotope ratios and Mg/Ca has been discussed separately by Bizzarro et al. (2023, *Astrophys. J. Lett.*, vol.) and is not from this study, so we have specified it clearly. The linear relationship is shown in the figure below (Bizzarro et al., 2023, *Astrophys. J. Lett.*, vol. 958. The figure below denotes $\delta^{25}\text{Mg}$ in ppm because of their very high accuracy of the measurements). Since we actually measured Mg-Fe carbonate breunnerite, which is also one of the major Mg-containing carbonate minerals in Ryugu samples, and therefore discuss $\delta^{25}\text{Mg}$ in bulk and leaching solutions, we have organized this discussion by clarifying the differences between the two studies. The plot of Ca/Mg versus $\delta^{25}\text{Mg}$ and the fraction containing carbonate has been added and discussed as Figure 5.

[REDACTED]

L305: even if any of this made sense, what is the uncertainty associated with the dolomite endmember composition? An isotopic difference of 0.3 to 0.4 permil is quite small, can you be sure that it is significant?

→ With regard to the linearity between $\delta^{25}\text{Mg}$ and Mg/Ca presented by the high-precision isotopic analysis of several aggregate samples by one of our co-authors' published work (Bizzarro et al., 2023, *Astrophys. J. Lett.*, vol. 958), it is considered that representative values have been obtained due to the large amount of samples. However, the distinct linear equation uses the Mg/Ca ratio of dolomite from other study to calculate $\delta^{25}\text{Mg}$ of dolomite. For this Mg/Ca we also considered that this corresponded to an uncertainty, as the representative values for Ryugu as a whole were not well compiled. The value 0.911 of Bazi et al. (2022, *Earth Planets Space*, vol. 74), applied by Bizzarro et al. (2023), was the most recent result at the time. There have been several follow-up reports since then, so in this study we have used the value of 0.948. The maximum difference between the calculated $\delta^{25}\text{Mg}$ values of the

endmember resulting from the use of the highest and lowest Mg/Ca of the published works is 0.07 permil, but we believe that this error factor can be reduced by using the average Mg/Ca value of published works. This corresponds to an error not much different from the analytical reproducibility of $\delta^{25}\text{Mg}$, but since the analytical reproducibility in our report is 0.05 permil and other studies have reported 0.02 permil or lower (e.g., Bizzarro et al., 2023). Although there is a 0.4 permil difference in $\delta^{25}\text{Mg}$ such, the difference contrast to analytical error is important. Therefore, we believe that the observed fractionation of a few tenths to over 10-fold is sufficient for discussion.

Another point is the selection of isotope fractionation factors, for which we have followed your advice and presented some models (Supplementary Figure).

L314: This was not observed. It was calculated based on assumptions.

→ We have added text so that it is clear that the values are calculated assuming isotopic fractionation.

L316: So phyllosilicates precipitate first, followed by dolomite and then more phyllosilicates? What evidence is there for such a precipitation order?

→ This precipitation order is derived from the fact that the difference between the $\delta^{25}\text{Mg}$ of the Ryugu dolomite and the fractionation value calculated from the experimental dolomite is well explained by the assumption that the phyllosilicate precipitates first in a closed system. Furthermore, the isotope ratios at the time of dolomite precipitation and at the last stage when water was present ($\delta^{25}\text{Mg}$ in the exchangeable pool) also show a decrease that could be due to continued precipitation of phyllosilicate. However, this is the simplest precipitation sequence and does not account for complex processes such as multiple precipitation events. Nevertheless, the $\delta^{13}\text{C}$ of the Ryugu dolomite is very homogeneous (Fujiya et al. 2023, Nat, Geosci) and does not appear to have been precipitated over a long period of time, thus dolomite precipitation environment did not change significantly.

L324: This is quite vague – do you mean that the carbonate leaches are isotopically heavier than expected? If so, state that here.

→ At the planning stage of this study, it was planned to selectively extract Mg from Ryugu carbonates with the weak acids EDTA and CH_3COOH , which are used for stage-leaching of terrestrial samples, but Ryugu particles are more reactive than the terrestrial materials, so the EDTA and CH_3COOH fractions were not solely composed of carbonates but with Mg from phyllosilicates. For this reason, a physical pick-up of the particles was added (i.e., single-mineralogy breunnerite grain, Figure 1). As the terminology is likely to be misleading, as

you have pointed out, we have standardized it as follows.

L336: What is the SOM leaching?

- We have added that this means the extraction protocol for soluble organic matter (SOM) presented by Naraoka et al. (2023, Science).

L334-352: This is a rambling discussion that doesn't really mention Mg. A lot of the information provided is not required, instead it would be better to focus on what Mg is doing and use the behavior of other elements to support the explanation.

- We agree with that the content is not essential to the discussion in this chapter, so we have moved most part of this paragraph to Supplemental Information. We have left only the statement regarding the relative abundance of magnesium.

L364: Is that the current consensus with regards to exchangeable Mg or is there debate in the literature?

- The isotopic fractionation of exchangeable pools is probably best experimentally validated by Wimpenny et al. (2014, Geochim. Cosmochim. Acta). In this study, representative clay minerals, illite, kaolinite, and montmorillonite, were used to verify degrees of Mg isotopic fractionation of solutions in contact with the exchangeable fraction. Although the ion exchange capacities (CEC) of these minerals vary widely, in all cases isotope fractionation was shown to be negligible.

More recently, Cai et al. (2022, Geochim. Cosmochim. Acta, vol.321) reported that regolith-exchangeable fractions and ambient groundwater of gneiss-covered catchments show similar Mg isotopic ratios. They also conducted batch adsorption/desorption experiments using collected regolith samples and have reported negligible Mg isotopic fractionation during adsorption/desorption.

In contrast, the exchangeable fractions of synthesized clay minerals have been reported to be " *$\delta^{26}\text{Mg}$ values of the exchangeable pool were lower than, or within error of, the initial solution*" (Hindshaw et al., 2018, Earth Planet. Sci. Lett., vol. 531). The structural Mg of their synthetic clay is enriched in ^{24}Mg , in contrast to natural reactions where phyllosilicates are typically enriched in heavier Mg (^{25}Mg and ^{26}Mg) relative to the unaltered silicate materials. Whether these artificially synthesized clays are representative of natural isotopic fractionation behavior needs to be tested in the future.

Structural Mg silicate residues in Ryugu phyllosilicates are also characterized by heavy Mg enrichment, and we applied the experimental results of Wimpenny et al. (2014) and Cao et al. (2022), who also used natural layered silicates.

L370: what are the other solutions? The progressive leaching removes exchangeable cations before attacking carbonates and finally any silicate materials. So chemical and isotopic differences reflect that different reservoirs are being targeted, rather than changes in the solution composition. You have tried to back out the composition of the aqueous phase during carbonate precipitation but this is not what is shown in the ternary diagram or the measured element ratios.

→ We agree with this point. It meant that the elemental composition and isotope ratios were different from those of the other fractions, but this overlaps with the discussion of the exchangeable fractions in this paragraph, so the text has been simplified.

L382: What do the arrows signify? I would have thought smaller ions (Li and Mg) are more likely to diffuse into the interlayer region of a clay mineral than large ions such as Ba.

→ The partition experiments are determined by repeated treatment of water dispersion with solutions of alkali metal chlorides and ammonia. It is generally known that the exchange reaction is selective for ions with larger ionic radii and valence state, which is probably related to the radius and dehydrated of hydrated cations. As for the interlayer exchange site, the hydrated ion must first approach close to the interlayer exchange site as a hydrated ion, so Mg with a large hydration radius probably be at a disadvantage for exchange reaction. As you pointed out, Mg^{2+} has the smallest ionic radius between major cations, but the radii of the hydrated ions are 4.76 angstroms for Mg^{2+} compared to 2.75, 2.32 and 2.95 angstroms for Na^+ , K^+ and Ca^{2+} , respectively. By volume, Mg^{2+} is 400 times larger than dehydrated Mg^{2+} , which is in contrast to the other major cations, which are 4 to 25 times larger (Table below, from Maguire & Cowan, 2002, *Biometals*, vol. 15). This high affinity of Mg^{2+} for water leads to a water exchange rate that is 3 to 4 orders of magnitude lower than that of the other major cations. The similar situation exists also for Li, but the large volume change and the strength of the interaction with hydrated water can be a barrier to the exchange reaction.

L388: How is this selective adsorption constrained and what is the mechanism?

→ Saponite, as well as other phyllosilicates, have several exchange centers in their structure, and the selectivity in the exchange of ions is related, for example, to the increased role of broken SiO links on the lateral faces and ribs of the saponite crystal (Polyakov and Yu. I. Tarasevich, 2012, *J. Superhard Mat.*, vol. 34).

L391: In many clay minerals, particularly expandable clays like smectite, the majority of exchangeable ions enter the interlayer to balance charge, rather than being chemically bound. These minerals have a far greater cation exchange capacity than non-expandable clays such as

kaolinite. Have you considered interlayer expansion and uptake of ions into this region? The later discussion of the dissolved composition is solely based on the surface layer exchange, which may be misleading.

→ My apologies for the misunderstanding, but the exchangeable pool components are extracted from both the surface and interlayer of the phyllosilicate, whereas the structurally exchanged sites are also present in the interlayer (e.g., Wimpenny et al., 2014). We have made changes to clarify these differences.

L397: Don't these partition coefficients essentially mean that Na will always dominate the cation pool no matter what the composition is? What are the average element ratios in the exchange pool? It would be useful to see these for some context.

L400-408: Again, this discussion should be simplified; do we really need all of these ratios? The problem is that the thrust of the discussion is lost. I am not sure why any of this is important.

L443: See earlier comment about the KD values used here – they would always select for Na preferentially in the modeled solution.

→ The ratios of the cations in the fluid of aqueous alteration were calculated using these values and the partition coefficients of Tipper et al. (2021, Proc. Natl. Acad. Sci. U.S.A., vol. 118). The text has been changed to make it clear that the ratios are the result of calculations using the partition coefficients from Tipper et al.

The fluid chemistry reconstruction of the Ryugu parent body has been discussed using a different approach, i.e., chemical equilibrium modeling based on mineral composition (Nakamura et al., 2023, Science, vol. 379), which also suggests that the initial fluid rich in Mg-Na-Cl changed to a composition rich in Na-Cl. Our results could be used as a constraint for key parameters of aqueous alteration such as W/R and pH. The text has been revised to make it clear that the results are useful for cross-validation of different scientific approaches and as tie points for aqueous alteration reactions.

L398: ...using the average element ratios...

→ We have changed as suggested.

L412: Is this a surprise? Na has long been known to be very mobile during weathering.

→ While this has been verified by chemical equilibrium modeling for water quality (Zolotov et al., 2012, Icarus, vol. 220; Nakamura et al., 2023, vol. 379) and is predictable given the general properties of the elements, we believe that the important thing is that the actual evidence has been corroborated using Ryugu samples that have not suffered any terrestrial weathering and alteration (mineral oxidation, reprecipitation of secondary minerals, such as

Mg sulfate, etc.). The conditions of aqueous alteration and solute compositions can be simulated under conditions of temperature, pressure, water/rock ratio (W/R), redox conditions, fluid movements, and gas escape (Zolotov et al., 2012, *Icarus*, vol. 220), which would provide Ryugu's reaction conditions consistent with the composition of dissolved elements obtained in this study. In addition, our previous study detected an unexpectedly large amount of organosulfur anions (Yoshimura et al., 2023, *Nat. Commun.*, vol. 14), which would need to be considered in conjunction with the cation composition and Mg partitioning conditions.

L426: The breunnerite grain is barely mentioned. To be honest, I had forgotten that you had analysed it.

L430: I don't understand this – the sequential leaching is the backbone of this paper! Was it not very good? In that case, what is the point of the paper?

→ As you pointed out, the discussion is weighted more toward the chemical composition of the leacheates than breunnerite, so I have added the summary from the sequential leaching to the first part of this conclusion to properly present the results to the reader.

L435: This is the first mention of any Mg isotopic fractionation factor. This should be introduced and explained in the discussion.

→ We have added to the discussion the isotopic fractionation factor for phyllosilicates.

L437: See earlier comment – what is the evidence for continued phyllosilicate precipitation?

→ Since the Mg isotope ratios may have decreased further (-0.55 permil compared to the starting point) after the precipitation of dolomite (-0.38 permil), this difference can be attributed to the continued selective removal of heavy Mg isotope (^{25}Mg) by the precipitation of phyllosilicates. We have added text to the discussion explaining that the endpoint of this magnesium isotopic change is lower than for dolomite.

Reviewer #2 (Remarks to the Author):

This paper is investigating the alteration history of Ryugu as viewed by Mg isotopes. Using a step leaching sequence from most soluble to least soluble, the authors can have sequentially access to the Mg isotopic composition of the exchangeable ions, carbonates and silicates, and analysed as well a breunnerite grain. They show a progressive enrichment in ^{25}Mg from most soluble phases to least soluble phases. Based on a simple model of phyllosilicate-carbonate precipitation, they

showed a 3 steps alteration with about 70% of Mg uptake from phyllosilicate precipitation followed by mixed phyllosilicate-carbonate precipitation and finally a last step of phyllosilicates. Using the most soluble leachate, they estimate the composition of the final fluid composition, and better understand the behaviour of the different phases in solution (organic matter for instance).

Overall Impression

This paper is rather well written and propose an interesting approach for deciphering fluid alteration characteristics and conditions. Despite being most of the time well written, some sentences/paragraphs could be rephrased (and some sentences cut in half) to facilitate the reading. The current version is every now and then a bit hard to follow as figure references are not always correct, but the results and discussions are well illustrated by numerous figures.

The discussion is highly structured which help to a good understanding. However, some more details can and should be provided in order to be able to reproduce model presented in this article and have a broader discussion. I think it would beneficiate from a bit of careful work on clarifying points and correcting some mistakes. That being said, the content of the paper is interesting, and the community will beneficiate for this high-quality study. Therefore, I recommend publication in Nature communications after major revisions.

General comments

Model of phyllosilicate and carbonate precipitation should be better details to be reproducible. Further discussion about the interpretation of Mg-isotopic composition of leachate should be done (some details are already given in caption of Fig. 4).

You model only a narrow range for carbonate precipitation. You should explain in more details why? Are you expecting only a short range for dolomite/carbonate precipitation or could you envision a wider range but we only access to average due to the nature of analysis?

→ The duration of dolomite precipitation is discussed with reference to the constraints from oxygen-carbon isotope ratios, which Fujiya et al. (2023, Nat. Geosci., vol. 16) concluded that calcite was initially precipitated over a wide range of temperatures and oxygen partial pressures, whereas the ratio of CO₂/CO/CH₄ molecules in the gas changed over time, and the system approached equilibrium when the dolomite precipitated. A follow-up report states that this precipitation of dolomite occurred after retrograde cooling had progressed. Considering the homogeneity of oxygen and carbon isotope ratios of carbonates, we do not think it is reasonable to consider multiple precipitations or precipitations over a wide period of time where oxygen and carbon isotopes can change. Dolomite precipitation temperatures and ages are being constrained by other indicators (e.g., Yokoyama et al., 2023, Science; Fujiya et al., 2023; Nakamura et al., 2022).

Since the Mg isotopic ratio of carbonates shows about the history of Mg partitioning between secondary carbonates and phyllosilicate, we have followed the comments and presented a figure showing the conditions under which the Mg partitioning ratio was varied with respect to phyllosilicate (Supplementary Figure S6). The extremely early or late precipitation rates relative to phyllosilicate may lead to heterogeneity in Mg isotopic ratios among dolomite particles, but such a discussion of precipitation timing needs to be constructed in conjunction with inter-particle comparisons of oxygen and Mg isotopic ratios.

The data could be better synthesised at least in supplementary table and clearly identified with new data from this paper along with literature data necessary to the conclusions of this paper, also in figures (e.g., Fig. 2). Besides, I do not see real contribution of Table 1 in this study.

Reference to table or figure is sometimes wrong so that it can be complicated to really follow what the authors want to tell us. Please have a careful look to the reference to figure, figure panel and table.

Some references are cited as 2023 but are 2022. Please correct accordingly throughout the text. Some other references might also be 2022 and not 2023, please have a careful look on the references.

It is ok for the review, but lots of “–“ sign are not on the same line as the figures. Please be careful during the proof stage to that point.

→ Please refer to our responses to the comments we received on each of them individually.

Detailed comments (line numbers refer to the beginning of a sentence)

Main Text

Intro

L75: “radiative”, I think it is rather “radioactive”.

→ We have changed the word.

L81: Nakamura et al., 2023 might be 2022. Please check reference.

→ Regarding the year of publication of this paper, 2022 might be said to be correct. However, the official publication in *Science* is in a special volume with six other Ryugu papers, Volume 379, Issue 6634, published on 24 February 2023. Nakamura T. (2023) was accepted earlier than other papers, thus has been posted online for a longer period of time in 2022. There is also Nakamura E. et al., 2022, by the same family name person, which we cite in the Supplementary Information. In the main text, we refer to Nakamura T. et al. (2023).

L91: “see Table 1 for carbon abundances and isotopic compositions”. I do not think this is the

appropriate place for this reference. You speak about mineralogy. Table 1 is volatile content and isotopic composition. Also, clarify the input of data in table 1 for this study. Such data are not really discussed in the text.

→ We agree with this point and have removed the compositions that are not relevant to the main discussion and moved them to Supplemental Information.

L94: “breunnerite [(Mg, Fe, Mn) (CO₃)₂].” I think there is only 1 CO₃. Please correct accordingly.

→ We have corrected this point.

L120: “Given that the isotope fractionation factors (α) for carbonate ($\alpha < 1$) and clay ($\alpha > 1$) minerals are generally opposite in sign (Saenger and Wang, 2014; Wimpenny et al., 2010, 2014), the $\delta^{25}\text{Mg}$ composition of the residual liquid phase is expected to differ from that of the starting solution according to which mineral precipitates first.”

o Please at “to” between expected and differ (bold).

o I am not sure that I understand the implication between the two part of the sentence. In any case, if there is isotope fractionation of a single species, the residual liquid phase is expected to differ from that of the starting solution. Do you want to say that as there are two species involved with opposite fractionation factor, the evolution of the residual liquid phase will depend on which phase is precipitating first? I would also mention the evolution is function of the values of the fractionation factors and the proportion of each phases to precipitate. Please clarify this sentence.

→ As you suggested, we have revised the text:

“One major difference between major secondary minerals is that the isotope fractionation factors (α) for carbonate ($\alpha < 1$) and clay ($\alpha > 1$) minerals are generally opposite in sign (Saenger and Wang, 2014; Wimpenny et al., 2010, 2014). As there are two species involved with opposite fractionation factor, the isotopic evolution of the residual liquid phase will depend on (1) which phase is precipitating first, (2) the values of the fractionation factors, and (3) the proportion of each phase that precipitates.”

L149: it if Fig 2C. I will try to tell you the other one, but please check all your figure 2 references since you seemed to have added a panel to that figure without updated the references.

→ Thank you for pointing out the correction. We have corrected these errors.

L150: You describe breunnerite as [(Mg, Fe, Mn) (CO₃)], saying Mg, Fe and Mn are the main compounds of the crystal, but Ca is actually more abundant than Mn and not discussed (Fe/Ca = 2.45; Ca/Mn = 3.10). Why Ca is not in the formula? Could this grain actually be a small-scale mixture with calcite?

→ Breunnerite is generally represented as (Fe, Mg)CO₃ (e.g., according to Mindat database, breunnerite is “*Fe-bearing variety of magnesite with the Mg:Fe atomic ratio ranging from 90:10 to 70:30*”), but since it often contains manganese, we agreed on the notation (Fe, Mg, Mn)CO₃ with some co-authors of the Hayabusa2 initial analysis scientists. Since it is a rhombohedral crystal-like dolomite, it is thought that manganese, whose ionic radius is almost the same as that of Mg, is often substituted in natural products, and in fact it probably forms a solid solution with some endmembers. The breunnerite in this study is in good agreement with the peak of Urashima et al.'s breunnerite (see peak list below for reference), and no subpeak corresponding to calcite was observed. Laser Raman mapping analysis also shows that the calcite peaks in the particles are below the detection limit. Therefore, we believe that the crystal structure corresponds to breunnerite, and that the concentrations of Ca and Mn are most likely incorporated during the precipitation of breunnerite by substitution reactions in accordance with the concentration ratios of cations dissolved in water.

Previous electron microprobe analyses of breunnerite have also reported Ca equivalent to the amounts in this study. In addition to manganese and calcium, sodium was also detected, which may be an effect of partitioning from a solution of sodium-rich aqueous alteration. We did not include details of elemental concentrations because of our intention to focus the discussion on magnesium, so we follow the comments and show them below and in the supplement. In this case, a portion of the nitric acid solution in which the two particles were dissolved together was analyzed by ICP-MS, and most of the remainder was analyzed for Mg isotopes. Due to the very small sample size, the 2RSD of the measured signal is much larger than that of the other elements because the Ca concentration was performed for ⁴³Ca, which has a low abundance.

	mol/mol	2SD
Fe/Mg	0.319	0.021
Na/Mg	0.070	0.001
Ca/Mg	0.129	0.010
Mn/Mg	0.041	0.001

L150: Is breunnerite a super group that encompasses calcite, siderite, dolomite and rhodochrosite? It is not really clear when you say breunnerite because there is no breunnerite standard in the supplementary figure S2.

→ Breunnerite is a carbonate with the rhombohedral structure as dolomite and others. The comparison of peak positions for breunnerite is based on a comparison with the reference

paper, Urashima et al. (2022, Anal. Sci., vol. 38). The spectrum of breunnerite was not included in our laboratory's standard database, and therefore was not included in Supplementary Fig. S2. Their Raman spectra were measured with exactly the same instrument configuration and show good reproducibility of peak positions for two types of natural breunnerite. Please see the table below (the unit of peak positions T, L, ν_4 , ν_1 is cm^{-1}). Four main peaks are shown for each carbonate mineral.

	End member formulae	T	L	ν_4	ν_1
Calcite	CaCO_3	154.6 ± 0.9	280.9 ± 0.9	711.6 ± 0.8	1086.0 ± 0.8
Magnesite	MgCO_3	212.5 ± 1.0	329.3 ± 1.0	738.2 ± 1.1	1094.4 ± 1.0
Siderite	FeCO_3	181.9 ± 0.7	283.4 ± 0.5	728.9 ± 3.7	1084.7 ± 0.2
Rhodochrosite	MnCO_3	184.8 ± 0.4	289.9 ± 0.5	721.2 ± 0.5	1087.7 ± 0.2
Breunnerite (KP)	$(\text{Mg, Fe})\text{CO}_3$	204.8 ± 0.4	316.2 ± 0.4	734.8 ± 0.3	1092.2 ± 0.3
Breunnerite (NC)	$(\text{Mg, Fe})\text{CO}_3$	204.6 ± 0.4	316.3 ± 0.5	735.2 ± 0.4	1092.7 ± 0.5
Dolomite (Azc)	$\text{CaMg}(\text{CO}_3)_2$	176.7 ± 0.5	300.2 ± 0.8	725.2 ± 0.4	1098.5 ± 0.5
Dolomite (BC)	$\text{CaMg}(\text{CO}_3)_2$	176.7 ± 0.4	300.2 ± 0.4	724.5 ± 0.7	1097.8 ± 0.9
Dolomite (LF)	$\text{CaMg}(\text{CO}_3)_2$	176.4 ± 0.2	299.6 ± 0.3	724.3 ± 0.7	1097.4 ± 0.7
Dolomite (Bin)	$\text{CaMg}(\text{CO}_3)_2$	176.9 ± 0.4	300.6 ± 0.7	725.6 ± 0.2	1099.1 ± 0.2

L150: Can you provide the XRF spectra, at least in the supplementary material, as well as elemental compositions for the breunnerite.

→ Please refer to the XRF data as shown below. The blue color shows Ryugu's breunnerite particles, and the orange color shows calcium carbonate analyzed under the same conditions as a reference. Actually, the XRF analysis has a large probe size, so the XRF data also includes X-ray fluorescence from the surrounding area of the particles. This may cause a misunderstanding that signals such as Ti and Cl originated from the Ryugu carbonate grains, so we did not show this data in the submitted manuscript. The clear peaks of Mg and Fe, which are not found in the reference carbonate, support that the mineralogy is breunnerite, as do the ICP-MS analysis and Raman spectra.

L153: “(Figs. 2C and 2DP).” Please delete the “P”. Update reference: 2D and 2E.

L155: Update reference: 2D.

L156: Update reference: 2E.

→ We have corrected these errors.

L154: Can you give extend of peak position? It is hard to really read that information from the figure. What is the precision (reproducibility) you have on the position of the peak?

→ In fact, due to contractual issues with the company with whom we collaborated on this Laser Raman analysis, we do not have access to the raw data, so we only have the uneditable figures and photos. We confirmed peak positions in their lab and determined mineral

compositions on site. We have added the peak position of breunnerite from Urashima et al. (2022, from Table above) and the new elemental ratios measured by ICP-MS to show that it is definitely breunnerite.

L157: It seems that the variation of the peak position for the breunnerite peak and the carbonate peak are independent. Can the surface irregularities affect the position of both peaks differently?

→ As you pointed out, some of the peaks are independent, and some crystal structures produce a cation-dependent trend. The width of the peak position indicated by the color tone of the contours is quite small compared to the difference in peak position for each mineral. In Urashima et al. (2022), the peak features used for this Figure 2 mapping are reported in detail for each carbonate mineral species, with vibrational frequencies plotted biaxially to visualize the cation-dependent frequency shifts (cited below). Importantly, in the biaxial plots of ν_1 and L wavenumbers (Figure below), each mineral is well separated. In the dolomite series, cation substitution causes a high-frequency shift in the order $\text{Mn}^{2+} < \text{Fe}^{2+} < \text{Mg}^{2+}$, but the peak position of the breunnerite spectrum obtained in this study is in good agreement with that of Urashima et al. (2022), so the substitution of other cations, replied to the comment on L150, no effect from incorporation of Mn, Ca, or Na into breunnerite was observed. The lower figure also shows that the data points of breunnerite forms a cluster completely different from other carbonate minerals.

[REDACTED]

L158: Please add “quadrupole” before inductively coupled plasma mass spectrometry (ICP-MS)

as in the material and method part.

→ We have added “quadrupole”.

L160: I am not sure “whereas” is needed here.

→ We have deleted, and also added that Ca and Na present in breunnerite.

L164: For non-aware reader, this paragraph could be hard to follow because they do not know what is extracted by each solvent. It could be a good idea to summarize what is expected (maybe with the help of Supplementary Table S2).

→ Supplement Figure S1 also summarized the minerals that were targeted for extraction, so we added the text to summarize and direct the reader to Supplementary Fig. S1 and Table S2. We also added a description and Ca/Mg- $\delta^{25}\text{Mg}$ plot as Figure 5.

“The EDTA fraction targets carbonates that are more soluble in acid than in CH_3COOH (e.g., the former is used for selective extraction of calcite, and the latter for selective extraction of dolomite, see Supplementary Fig. S1). The plot of Ca/Mg versus $\delta^{25}\text{Mg}$ (Fig. 5) suggests that there is a contribution from the breunnerite component in the EDTA fraction. In contrast, the CH_3COOH fraction plots on a mixing line between the total digests of the Ryugu samples and dolomite; this result suggests a significant contribution mainly from dolomite to the CH_3COOH fraction. However, because CH_3COOH also partially dissolves phyllosilicates (Supplementary Table S1), Mg derived from phyllosilicates was also present in that fraction (Fig. 5). The HCl fractions are characterized by higher $\delta^{25}\text{Mg}$ and lower Ca/Mg compared with the four bulk digests of Ryugu samples (Bizzarro et al., 2023). The compositions of the sequential extracts with carbonate (dolomite and breunnerite) and phyllosilicate are used as mixing endmembers of Mg. The HF/ HClO_4 fractions are not shown on the Ca/Mg plot because Ca was at the lower limit of detection.”

L178: “see Supplementary Fig. S3 for the $\delta^{26}\text{Mg}^*$ value in each leachate in this study and in the SOM leachates reported by Yoshimura et al., 2023).” I would rather do a sentence out of the brackets saying that you observe not 26Mg-excess in the leachate. E.g., “ $\delta^{26}\text{Mg}^*$ value in each leachate in this study and in the SOM leachates reported by Yoshimura et al., 2023 are indistinguishable from 0 ‰ within errors (Supplementary Fig. S3)”. Or “No 26Mg-excess is observed in each leachate in this study and in the SOM leachates reported by Yoshimura et al., 2023 (Supplementary Fig. S3)”. Actually you have a good explanation in supplementary material but I would nonetheless do a one sentence summary of that in the main text.

→ We have added the text you suggested without parentheses.

L198: It is not clear which are “these extracts”. I would put back the names of extracts you mention.

→ We have changed the text to “EDTA, CH₃COOH, and HF/HClO₄”.

L201: “high Mg/Ca”. Is it high in general or high in comparison with the other? Because it seems to be pretty similar compared to H₂O and NH₄Cl. Please correct accordingly. Mg/Fe is intermediate. Please give more details about what you mean.

→ We have clarified which extracts Mg/Ca and Mg/Fe we are referring to, giving specific values.

L220: Yes, indeed $\delta^{26}\text{Mg}$ is affected by radioactive decay of ²⁶Al. However, the deviation should be on the order of 10s of ppm (Al/Mg ratios close to solar ratio or below), so well below the mass dependant variation you should observe on the permil/0.1 permil level. “However, ²⁶Mg is affected by the radioactive decay of ²⁶Al. Therefore, $\delta^{25}\text{Mg}$, which” I would rather note $\delta^{26}\text{Mg}$ to be consistent with the next sentence.

→ We have changed the text according to this comment.

L242: Do you mean supplementary figure S1? Please correct accordingly.

→ We have corrected the error.

L245: “which is as low as that of terrestrial carbonate precipitates”. Do you mean the deviation compare to the bulk is similar to terrestrial carbonate or the absolute value? If it is absolute value, what does this mean? Please clarify in the text.

→ The sentence was to emphasize that a ²⁵Mg/²⁴Mg fractionation exists as observed on Earth. As you pointed, we do not think it was necessary to contrast it with terrestrial observations, so we deleted the sentence.

L245: I am not sure about the meaning of the evolution trend. I agree that is an observation, but for further interpretation you would need to take into account fractionation factors, as you do in following part of the discussion. You can add a sentence to invite the reader to be cautious about direct interpretation.

→ A caution has been added regarding the influence of selection of isotopic fractionation factor and partition ratio of Mg into the dolomite and phyllosilicate. Also added a supplemental figure for the sensitivity of the model response to changes in these factors.

L247: Please also give error for the Mg isotopic composition. Please also clarify the number of

grains measured and if the reported value is the average of multiple measurements.

→ We have added 2SD value of 0.02 permil for this sample. The $\delta^{25}\text{Mg}$ value of breunnerite was obtained by dissolving the two grains shown in Fig. 2A together. The reason was that only one grain was not enough for the minimum amount of tens of nanograms required to measure the Mg isotope ratio. The 2SD values of each data were calculated from triplicate measurements of each sample.

L264: It is the first time you use $\Delta^{25}\text{Mg}$, please define it.

→ We have added an explanation to the text.

L267: chondrite, should be plural.

L281: “in the case of inorganic carbonates precipitated from solutions”, precipitated might be precipitation. Please read carefully this sentence and correct accordingly.

→ We have corrected these points accordingly.

L282: Please define “saturation indices Ω ” for non-specialised people.

→ We have added that an explanation for Ω , which is a saturation state relative to the stoichiometric solubility product, using calcite as an example.

L291: Do you think it would be possible to estimate temperature variation from $\delta^{25}\text{Mg}$ of breunnerite. I mean T dependency is low so you would need a good precision as well as a good estimate of the $\delta^{25}\text{Mg}$ of the fluid from which the breunnerite precipitate from.

→ It is likely that there is a temperature dependence of Mg isotope fractionation in breunnerite as similar in calcite and dolomite, but perhaps precipitation experiments under anaerobic conditions, where iron can be dissolved, are needed. This is still a future study.

L299: Why you are not showing a $\delta^{25}\text{Mg}$ and Mg/Ca plot? It seems to be an important plot to understand the data and your discussion. It might be a 2-panel figures along this your current Fig. 4.

→ As you pointed out, it is a very important figure, but it is not the original data for this discussion, and Bizzarro et al. presented it clearly, so we have tried to clarify the source in the citation.

L301: in supplementary table S3, please explicit what is “-“. No data available/calculated?

→ We have added that no data available.

L302: During the first reading, it is not clear how you calculate the theoretical dolomite data. You can add in bracket $\delta^{25}\text{Mg}_{\text{Dolomite}} = \Delta^{25}\text{Mg}_{\text{dolomite-aq}} + \delta^{25}\text{Mg}_{\text{Fluid}}$. But it should also be clearer if you define $\Delta^{25}\text{Mg}$ when using it the first time as said in a previous comment. Actually, some details are given in the figure 4 caption. It should be also mentioned clearly in the main text. Besides “Theoretical dolomite” in Fig. 4 is not explicitly denoting that it is theoretical in a way of early precipitation if nothing else occurs before that. You will gain to clarify this whole paragraph by better explicating the situation you consider.

L304: Please give here $\alpha/\Delta^{25}\text{Mg}_{\text{Phyllosilicates-aq}}$ for phyllosilicates ($\Delta^{25}\text{Mg}_{\text{Phyllosilicates-aq}} = +0.28\text{‰}$ from Fig. 6).

→ We have changed the text accordingly.

L309: Can you precise what do you mean by Mg partitioning ratio of 9:1. Is it the total partitioning of Mg between phyllosilicates and dolomite, or during precipitation of dolomite?

→ The Mg abundance in dolomite is determined by Bizzarro et al. (2023, *Astrophys. J. Lett.*, vol. 958) from a precise analysis of Mg isotopic ratios in several Ryugu aggregate samples, showing that up to 10% of the total Mg content in the samples is present as dolomite. We have modified the statement. Another Mg reservoir is phyllosilicate, that is altered silicate minerals, composed mainly of saponite.

L310: During the first read, it was not clear what was the cause and consequence. Maybe you want to say you need -0.38‰ decrease of $\delta^{25}\text{Mg}$, which using a simple model leads to 70 % of phyllosilicates before carbonate precipitation.

→ This is absolutely correct, and we have clarified it accordingly.

Does the model predict 70 % of phyllosilicates before carbonates or do you need 70 % of phyllosilicate to be product so that it matched the isotopic compositions? It is not clear what is assumed and what is deduce. Please clarify.

→ We have clarified tie points for this calculations based on observations and an explanation of the conditions assumed. The comments on the conditions for the amount of Mg distributed in dolomite and phyllosilicate were very important, so we have presented the new calculation results as Figure 7B and supplement figures. The influence of various conditions on the calculation of Mg distribution has been clarified.

L311: Besides, it is more about 75 % than 70 (74.3 % from my calculation). Please check also the Fig. 6.

→ The intent of the text was that precipitation would begin at about 70%, so it was changed to

make it clear.

L317: The offset of -0.55 ‰ from the first extraction to the bulk lead to only 90% of Mg 'loss'. What happen to the remaining 10 % of Mg? Besides, I think you need to compare the Mg isotopic composition of the fluid, not of the extraction, so you would need to correct for the $\Delta^{25}\text{Mg}_{\text{phyllo-aq}}$. Trying to reproduce the same model, accounting for $\Delta^{25}\text{Mg}_{\text{phyllo-aq}}$, this leads to an offset of -0.83 ‰ which would correspond to about 98.2/95.6 % of Mg uptake, much better than the current 90 % (depending on if I let dolomite precipitation until the end or not). Please correct and clarify this point.

Besides, you only tell L362 that the uptake of exchangeable would occur without fractionation. You should tell that here. If true, then I come back to my first sentence, what about the last 10 % of Mg? Do you have to change the fractionation factor for phyllosilicate, which would thus delay the precipitation of carbonate to a further extend of Mg uptake before? I do not think there are enough details about the model. Please give more information about how you define carbonate precipitation (end of the precipitation?), what correspond exactly the partitioning of Mg phyllosilicate:carbonate = 9:1 (during carbonate precipitation or in total).

→ This misunderstanding is due to our incomplete description of the figure caption as well as in main text. This figure shows the 10% carbonate contribution and the 90% silicate contribution overlaid on top of each other. Thus the orange area is the section where 10% each of carbonate and phyllosilicate were removed at the same time, which means that 100% of the dissolved Mg was removed in total when added. We have added an explanation for this.

The four constraints based on Ryugu's analysis used in this model calculation are (i) the isotopic ratio of Ryugu's bulk (as the isotopic composition of the starting solution), (ii) the isotopic ratio of the Ryugu's dolomite, (iii) the isotopic ratio of the exchange fraction (the isotopic composition at the endpoint), and that (iv) up to 10% of the Mg in the Ryugu sample being present as dolomite (Bizzarro et al., 2023). The Mg isotopic fractionation factor for dolomite precipitation was taken from Li et al. (2015), but this is presented as a new supplement figure because, as commented by reviewer 1, other values should also be considered. Since the isotopic fractionation of Mg structurally incorporated in clay minerals is a calculated value set to satisfy the isotopic fractionation of a closed system from these conditions, we believe it is important to scrutinize the other conditions. According to the Reviewer's insightful comment that there should be conditions for lithologies with less Mg in dolomite, we have made such calculations as a sensitivity test, please see the supplement figure.

For the fractionation of phyllosilicates, the calculations are passively calculated so that the starting solution varies from 0 per mil to -0.55 per mil, and as mentioned earlier, no reservoir of 10% residual Mg is assumed, so the calculations are left as they are to maintain a closed system. We have also added a calculation to make the Mg of dolomite and phyllosilicate 5:95, as you pointed out, and the fractionation factor for phyllosilicate is a bit lower for this.

L322: Please cite Fig. 4 here in this paragraph. Are all points from leachates consistent with a carbonate endmember with a $\delta^{25}\text{Mg}$ of about -1.4 ‰? Is the breunnerite point consistent with such a model?

→ We have added Figure 4 as a citation. The impact from sources other than breunnerite and endmember is related to our response to L330, so please see there for more details.

L328: Do you have any idea about the cause of the higher reactivity of phyllosilicates in Ryugu compared to terrestrial one?

→ High reactivity is probably due to the fact that the contact with water and exposure to gases has been limited compared to terrestrial materials, and that they are very porous (samples are fluffy) and have a large surface area.

L330: You can discuss more the result from the leachates. Details are given in caption of Fig.4 but not really discussed more in the main text. For C0107 CH_3COOH : how you can distinguish between partial dissolution of labile phyllosilicates (fractionation of Mg/Fe) vs. mixing line with a more ^{24}Mg -rich carbonate? What would be the evolution during phyllosilicate precipitation? What about EDTA points higher than the dolomite endmembers (for both A0106 and C0107)? It is also only possible to explain these points by Mg/Fe fractionation or should these require carbonate endmember with less ^{24}Mg -enrichments, i.e., formed earlier in alteration history? Partitioning of Mg^{2+} in cation exchange pools and solute compositions

→ The endmembers of the hydrofluoric and perchloric acid mixture are all dissolved silicate residues, but apart from this we have also done a step extraction of phyllosilicates with HCl, and these have a Mg/Fe of 1.5 mol/mol and a $\delta^{25}\text{Mg}$ of about -0.17 ‰. If these components contribute, the endpoint may be slightly to the right. As for Mg/Fe changes, if we set an endpoint where dolomite and breunnerite are mixed, they will plot on the mixing line of the endpoint with the silicate residue (see reference figure below). We have added this possibility.

Since the carbonates formed in the early stages of water formation are mainly calcite, the Mg content is low and its abundance is not very large, so its contribution is considered small. Dolomite was formed during retrograde cooling and precipitated at a temperature of 37°C,

with a very homogeneous carbon isotope ratio and relatively stable chemical conditions.

L362: “From the cation exchange pool of phyllosilicates (e.g., surface sites and exchangeable interlayer sites), isotopic tracers have shown that dissolved Mg is taken up without fractionation (Wimpenny et al., 2014)” Should be said earlier? (Cf comment L317).

→ We have added a description of the fractionation behavior of exchangeable pool, and also clarified that the Mg isotope fractionation behavior differs from structural Mg in secondary phyllosilicates.

L369: “which has higher $\delta^{25}\text{Mg}$ than the olivine average (Deng et al., 2021)”. I would be cautious in that comparison as Deng et al. measured olivine from chondrules. I think it is not clear whether olivine from CIs and Ryugu are all derived from chondrules material or not. In the latter case, we do not really know the isotopic composition of such olivine.

→ We have removed the description according to your suggestion.

Conclusion

L435: Fractionation factors are inverted compared to their respective phase.

→ The error has been corrected and the value of the fractionation factor for phyllosilicate, which takes into account the distribution ratio of Mg, has been added.

L471: You speak about the largest single grain of breunnerite, in singular form whereas you show 2 breunnerite in Fig. 2. What about the second one? Should be plural in the text?

→ We have changed to the plural.

L480: What would be the reproducibility of the peak position?

→ The standard deviation value for two times measurements of the peak frequency is generally smaller than $\sim 2 \text{ cm}^{-1}$.

L485: These standards (breunnerite, ankerite, and kutnohorite) were measured or only compared from other studies? If you have measured them, please add them in Supplementary Figure S2.

→ We have clarified that the peak positions of these minerals were done by comparison with the cited literature.

Aggregate samples and sequential leaching

L498: Does the quantity of solvent have an effect on leaching? You have used the same amount of solvent when the amount of sample is multiplied by about 2.5.

→ Some data from experiments with different ratios of sample to extractant are shown in Supplement Table S2. In the case of hydrochloric acid, there is no problem when the amount of sample is smaller than 600 uL of reagent for 15 mg of sample, but when the amount is 600 uL for 75 mg of sample, the reaction does not proceed fully. Carbonates react more readily than clay minerals (bottom of table). Since we used a smaller amount of solid sample in this study (due to the allocated sample amount is very limited), we believe that the reaction is well completed. As for exchangeable ions, test experiments in our laboratory have confirmed that the linearity of Cation Exchange Capacity (CEC) can be maintained at 2.5mg-25mg of solids per 500uL.

L493: You give the exposure age of sample from chamber A. Are you expecting cosmogenic effects?

→ This description is presented as basic information on the production of the sample, and is intended to indicate that chamber C was calculated from the lower part of the chamber, which is less affected by surface exposure. The text has been changed to clearly state the intent.

Inductively coupled plasma mass spectrometry

L560: Can you precise in which lab the Mg isotope analyses have been performed?

→ We have added the laboratory information: Geological Survey of Japan (GSJ), National Institute of Advanced Industrial Science and Technology (AIST).

L565: Do the data reported be single analysis or did you do multiple replicates?

Availability of data and materials

Does the Hayabusa2 Science Data Archives will contain your data? Otherwise, will your data be available in a repository or only as supplementary material from this paper?

→ If accepted, we will include a table of data in the supplement for the data presented in the paper, and also make the raw data for the individual figure plots available as a source data file. The Hayabusa 2 data repository shall also be made available and clearly stated in the paper.

References

Please note that the three following papers cited from the main text and the last one from the supplementary material are cited as submitted. Please add the correct references when they will be accepted and published.

o Bizzarro, M. et al. 2023. The magnesium isotope composition of samples returned from asteroid Ryugu. *Astrophysical Journal Letters*. (submitted)

- o Naraoka, H. et al. 2023b. Hydrogen isotope compositions of the Ryugu sample and carbonaceous chondrites: Implications for origins of hydrous asteroids. *Meteoritics & Planetary Science*. (submitted)
- o Takano, Y. et al. 2023. Primordial aqueous alterations recorded in water-soluble organic molecules from the carbonaceous asteroid (162173) Ryugu. (submitted)
- o Yada, T. et al. A curation for uncontaminated Hayabusa2-returned samples in Extraterrestrial Curation Center of JAXA: From the beginning to nowadays. *Earth Planets Space* (submitted). These papers are 2022, not 2023. Please check carefully the references for possible other year mismatched.
- o “Yokoyama, T. et al. 2023. Samples returned from the asteroid Ryugu are similar to Ivuna-type carbonaceous meteorites. *Science* 379, eabn7850.”
- o “Nakamura, T. et al. 2023. Formation and evolution of carbonaceous asteroid Ryugu: Direct evidence from returned samples. *Science*, eabn8671.”

→ The citation information for Bizzarro and Yada has been updated since they have already been published.

As previously replied, Yokoyama and Nakamura have been first-online published in 2022, but the special volume was published together in Feb. 2023. Confusingly, the publication year in Google Scholar is unchanged from the first online publication in 2022.

Discussion on hydrogen data by Naraoka was not be essential for the discussion, so this reference has been removed.

Takano et al. have almost completed the peer review process for *Nature Communications* and are close to final acceptance. We will add publication numbers and other information when they are assigned.

Supplementary material

List of Abbreviations: I would do a list rather than a paragraph for clarity.

→ We have changed to Excel list.

L140: Fractionation factors for phyllosilicates are not consistent between main text and supplementary material: 1.00028 in main text and 1.00054 in supplementary material. Which one have you used? Please correct accordingly. Besides, if the first one is correct, it is then quite different compared to Wimpenny et al., 2014. Can you comment? Besides, give more details about how you estimate these two values.

→ Since most of the earth samples were reported using $^{26}\text{Mg}/^{24}\text{Mg}$, and $^{25}\text{Mg}/^{24}\text{Mg}$ was used in this study to ignore the effects of ^{26}Al necrosis, the fractionation factors were converted to make these intercomparisons. We have clarified this in the text.

L172: “ β value of 0.511”. This beta is kinetic. 0.521 is for equilibrium (Davis et al., 2015). Please correct the text with the good value. Have you used 0.521 for calculating the $\delta^{26}\text{Mg}^*$?

→ In the manuscript we submitted, we used 0.511 for the calculation because this will likely result in a positive $\delta^{26}\text{Mg}^*$ if natural fractionation is driven by equilibrium processes. We have corrected the sentence.

L198: “adsorptive”, is that correct?

→ We have corrected.

Supplementary table S1: You can delete the 2nd header of the table. There is no need. Or you need to change “Na+K” to only “K” for the 2nd header? I would add ratios discussed in “Ionic composition of the fluid in contact with cation exchange pools” part as well as average and reconstructed fluid compositions. I think this table is pretty important and could be also insert in the main text.

→ The second header needed to be changed to K instead of Na+K, so I have corrected it. Thank you for pointing this out. Since the main text is limited by the number of figures and tables, I have added these figures to the results in the main text.

Supplementary Figure S2: You misspell the name of the orange spectrum rhodochrosite. It misses an “h”. Besides, why you do not observe the features above 1200 cm^{-1} compared to the Caltech reference?

→ We have corrected the mineral name.

According to previous reports of Raman spectroscopy of carbonates, there are four strong bands in the Raman spectrum, and the vibrational frequencies of these bands are known to be about 200, 300, 700, and 1100 cm^{-1} , respectively, so it is possible to identify the mineral composition with them (Urashima et al., 2022, Anal. Sci., vol. 38). Urashima et al. (2022) does not provide data for wavenumbers above 1200, so we did not consider them in the identification. The difference in the peaks in this region may be due to impurities in the mineral specimen, but we do not know the details because the spectra were measured in other laboratories.

Supplementary Figure S3: Can you add $\delta^{26}\text{Mg}^*$ for Ruygu from Bizzarro et al., 2023? Can you also add the breunnerite sample?

→ We have added data for Bizzarro. Unfortunately, we could not include breunnerite in the plot with Mg/Al because the aluminum was below the detection limit.

Tables

Table 1: Data in this table are not really discussed in the text. What is the interest of C, N and H content on bulk samples for the step leaching?

→ These concentration and isotopic ratio information was included as a preliminary step in the step extraction to inform the reader that the chemical composition of samples A0106 and C0107 was close to the average Ryugu value. The following text has been added to clarify this:

“The abundances and isotopic compositions of C, N and H using tens to hundreds of grams of the A0106 and C0107 samples used in this study are similar to previous analyses of CI chondrites from Ivuna and Orgueil (Table 1). Therefore, the influence of lithological heterogeneity on the representativeness of the chemically extracted values is expected to be small. Other values measured for CM2 Murchison, Murray, Aguas Zarcas, CI Orgueil and C2-ung Tarda at similar weight scales are shown for comparison of chemical composition.”

Table 2: Can you calculate the Mg/(Mg+Fe) for breunnerite. I would add in the title of the table “and breunnerite”, as it is not really an extraction.

→ We have added 0.76 to Mg/Mg+Fe. Also changed the title as you suggested.

Have you measured the Mg/Al ratio for the breunnerite? It would be also interesting to plot it in the Supplementary Figure S3, if measured.

→ According to the ICP-MS measurements, the Al signal was within the variability of the blank solution and unfortunately could not be quantified.

In caption of Fig. 2, it is written “Then, a high-precision analysis of the magnesium (^{24}Mg , ^{25}Mg , ^{26}Mg) isotope systematics of the two grains was carried out.” It is not clear if you have measured separately the two grains or not. Please add details in the text and put the 2 values in Table 2 if you measured them separately.

I would add in this table the content of the Supplementary Table S1, as it is the subject of one part of the discussion.

→ The two particles shown in Figure 2a were dissolved together in solution and subjected to elemental ratio analysis by ICP-MS, purification and isotope analysis of Mg. As mentioned earlier, the reason for this is that the amount of Mg in one particle was not sufficient for isotope analysis. We have added this description to the main text.

Figures

Fig. 2: Can you remove the thin grey contour in panel D and E?

→ We have removed it from the Fig. 2.

Please add Orgueil data from Fig. 3 (cf comment of Fig. 3).

→ The data for the acid digests of Orgueil correspond to the CI plots on the upper right, so a note has been added to the caption.

Fig. 3 (L674): “The yellow circle with the cross represent the data from the #5 H₂O extract of CI-group Orgueil; these data are consistent with the #7 hot H₂O data (Supplementary Figure S1).”

I do not understand which is which. Why the H₂O extract is in the hot H₂O extract? Please also give the value for Orgueil as a comparison in Table 2.

→ We clarified, citing Supplementary Figure S1, that Orgueil #5 and #7-1 are separate extracts from the Soluble Organic Matter (SOM) extract. The difference between the two is whether they are room temperature water extractions or nitrogen purged to 105°C. I added that the significance of this data is that the Mg isotopic ratios are almost identical, suggesting that most of the Mg-containing phases can be dissolved in water at room temperature. We have also compiled the SOM extract of Orgueil's Mg isotopic ratios with the published ones, and added it as a new Supplementary Table.

Why not add the breunnerite(s) data in the Fig. 3? I would be nice to have a figure with all data, and you can easily extend the range of Fig. 3.

→ In Figure 3, the discussion was focused on leaching solutions, so breunnerite was excluded. This is also the reason that in Figure 4, which follows, we intended to focus our discussion on carbonates.

There are also blue-green points for CMs and CVs. Please correct the caption. Why a special highlight to the Bizarro et al., 2023 data?

→ Since the data from Bizarro et al. were directly compared to the Ryugu and meteorite samples in the initial analysis project of Hayabusa2, we highlighted them because the isotopic compositions are firmly comparable to each other (the same analytical conditions). In fact, I was wondering whether the data published by the group at the University of Copenhagen should be treated in the same way, but I highlighted only Bizarro et al. from the viewpoint mentioned earlier.

You can add an arrow to indicate increase in solubility/order of precipitation to help the reader, e.g., to support discussion L245. Please indicate with more detail what is extracted for carbonate.

You give such detail in supplementary figure S1.

Please clearly differentiate your data from previous data (e.g., using “*” for previous data).

→ We have added arrows and cited Supplement Figure S1 for guidance. Also added details of the cited data.

Fig. 4. Why not show the H₂O and NH₄Cl step also?

“Mixing curves for the residue and the carbonate end components are shown in gray and red, respectively.” Not sure to understand what you mean. The gray mining curve is also heading toward carbonate composition but of breunnerite.

→ Since the main purpose of Figure 4 was to discuss the phyllosilicate fractions to EDTA and CH₃COOH, which are carbonate-targeted fractions, the plot was intended to focus on the residues and carbonate components that are most likely to contribute to these fractions. The H₂O and NH₄Cl fractions are very small as Mg reservoirs, and some of the Mg/Fe variations are large, so the horizontal axis had to be extended considerably to plot them, so the discussion of the main components was given priority. Also, we have changed the sentence: “Mixing curves of the calculated dolomite end component with the residual fraction and the mixing curve of the breunnerite with the residual fraction are shown in red and gray, respectively.”

You should add a Mg/Ca plot as it is also discussed in the text. This would be a great support for the discussion.

→ Thanks for your very insightful suggestions. Following your suggestion, we have created a figure plotted as the inverse of the former to clarify the relationship between Mg/Ca and $\delta^{25}\text{Mg}$ and the end components. First, in the relationship between Ca/Mg and $\delta^{25}\text{Mg}$ to the linear equation already reported by Bizzarro et al. (2023), the HCOOH fraction reported by Yoshimura et al. (2023) was closest to the HCl fraction, reaffirming the influence of phyllosilicates. In this study, we also attempted to separate carbonate in two consecutive fractions, EDTA and CH₃COOH, and found that EDTA plots in a position where the influence of breunnerite is more significant, while CH₃COOH plots almost directly above the linear equation for dolomite and phyllosilicate. The CH₃COOH plots almost directly above the linear equations for dolomite and phyllosilicate. This may indicate a stepwise extraction of carbonates of different solubility levels. In light of this, we have changed the text to argue for the usefulness of the particle picking approach and the end component estimation approach of Bizzarro et al. (2023).

Besides, for simplifying the legend, I would use a symbol legend with only solvent name and a colour legend with Ryugu grains name.

→ We have changed it as suggested.

Please precise to what correspond the number on the mixing line: fraction of Mg coming from carbonate or fraction of carbonate compared to phyllosilicate (that is to say taking already into account the 9:1 ratio between phyllosilicates and carbonates)?

→ In Figure 4, only the mixing of carbonate and silicate due to dissolution is considered. In Figure 6, we have clarified that isotopic fractionation is taken into account to calculate the composition of the solution and the distribution of Mg to secondary minerals in aqueous alteration.

L708: Would breunnerite fit in the order of fractionation factors? What is the order of variations between the fractionation factor for the cited minerals?

→ For breunnerite, there is no experimental evidence of a fractionation factor with solution. This needs to be verified by creating an anaerobic solution in which both iron and magnesium can dissolve, for example, but this will be determined in the future, as there are

reports of experimental precipitation of siderite, although few have been reported. In fact, we are in the process of initiating such an attempt in our laboratory. In the magnesium isotope fractionation of carbonates, the order of the difference from solution increases in the order of dolomite, magnesite, and calcite.

Fig. 6. As also mentioned in the text, please better explain the model and how you define carbonate precipitation, here, in the main text or in the supplementary material. Please verify the extend of each region (cf comment L311).

→ Please see our response to L317, and we have added each constraint condition to the text. We have also added the calculations to see how each factor is affected by what you have suggested, so I have added this as a Figure 7B and Supplement Figure S6.

Reviewer #3 (Remarks to the Author):

This manuscript presents Mg isotopic compositions of leachates from asteroid Ryugu to investigate its aqueous alteration history. The authors observe a significant variation of the Mg isotopic compositions between the different leachates. The results show an enrichment in the lighter Mg isotopes in the breunnerite grains, the leachates with weak acid show less enrichment and the leachates with the strongest acid are the most depleted in lighter isotopes of Mg and corresponds to the bulk composition. From this data and input from the literature, the authors concluded that 70% of the Mg was precipitated into the phyllosilicates followed by dolomite precipitation and that the last solution that was involved in the alteration of Ryugu was Na-rich. While the introduction is well written and the objectives of the study are clear, I believe that the discussion part is hard to read and not well organised, this part would require some clarification. Furthermore, important data considered in this manuscript and, used in the interpretation of the results, are from Bizzarro et al., 2023 which is only a submitted work. Until this work is accepted, I am afraid that I can not advise on the publication of this paper. Two others cited papers are also only submitted work (Takano et al., 2023 and Naraoka et al., 2023) but their data are not used in this manuscript. Overall, the interpretation of the data seems accurate (though it is not my speciality) and worthy to be published to Nature Communications. Hence, I approve the submission of this paper if the discussion is re-written to make it more clear and straightforward and after the manuscript of Bizzarro et al., 2023 is accepted.

→ Bizzarro et al., which is important to the discussion of this study, has already been published by the *Astrophysical Journal Letters*, and the source of the data has been revised and clarified. The article by Naraoka et al. (2023b) has been deleted because its hydrogen content is less

relevant with the discussion. Takano et al. have almost completed the peer review process of Nature Communications and are close to final acceptance. We will add publication numbers and other information when they are assigned.

Line 57: Why using two different notation ($^{25}\text{Mg}/^{24}\text{Mg}$ and $\delta^{25}\text{Mg}$) to present Mg isotopic composition?

→ We have corrected the sentence according to the comments on L57 and L164:

“Breunnerite was the sample most enriched in light Mg isotopes, and the $^{25}\text{Mg}/^{24}\text{Mg}$ value of the fluid had shifted lower by $\sim 0.38\%$ than the initial value (set to 0‰) before dolomite precipitation.”

Line 123: “the ^{25}Mg composition of the residual liquid phase is expected differ from that” do the author mean “is expected to differ”?

→ We have made the following modifications along with the addition of explanations:

“Because there are two species involved with opposite fractionation factors, the isotopic evolution of the residual liquid phase depends on (1) which phase precipitates first, (2) the values of the fractionation factors, and (3) the proportion of each precipitate.”

Line 164: “the $\delta^{25}\text{Mg}$ was highest” is a bit clumsy. I would say: The most enriched sample in heavy Mg isotopes is etc. (same comment for lines below)

→ We have corrected accordingly.

Line 178: “ $\delta^{26}\text{Mg}^*$ ” What does the * represents here? If radiogenic why present this without talking about it?

→ As you pointed out, we meant radiogenic, but it is indeed confusing and not directly related to what we are discussing here, so I removed it.

Line 286: I believe that in order for Mg isotopes to be an indicator of temperature, one need to know if the isotopic fractionation is under equilibrium or kinetic conditions, which is not the case in this study.

→ We have added a clarification in the text that as a thermometer, it must follow an isotope equilibrium reaction.

REVIEWERS' COMMENTS

Reviewer #1 (Remarks to the Author):

The authors have done a good job of revising the manuscript and provided an extensive response to my earlier comments. I believe it is in much better shape than the first draft and support publication.

A few minor comments below:

L172: '...with a d25Mg value of...'

L174: Again, you need to specify what the value is ('...with a d25Mg value of...')

L239: I assume you mean micrograms?

L337: You have not analyzed the dolomite endmember, but the way it is written makes it sound like you have. Make it clear that you are comparing the calculated endmember composition and a theoretical composition here.

L339: There must be some uncertainty associated with the theoretical d25Mg value. How large is this compared to the estimated 0.3-0.4 permil difference between calculated and theoretical compositions?

L349: Again, there is no observed isotopic composition of dolomite. It is either calculated from the mixing array in d25Mg vs Mg/Ca or is purely theoretical.

Reviewer #2 (Remarks to the Author):

Review in the attached pdf.

[Editorial Note: This file is displayed over the next 3 pages]

Review: NCOMMS-23-47707A – “Breunnerite grain and magnesium isotope chemistry within cation-partition dynamics during aqueous alteration of the asteroid Ryugu” by Yoshimura et al.

Overall impression

Thanks for having chosen the "transparent peer review", which I consider as an interesting approach for improving the quality of the manuscript. I also thank the authors for the detailed answers and the modifications in the new version of the paper. I think they have done a good job for improving the manuscript. However, I still have few comments that need to be addressed before publication.

General comments

1- Figure 7. The model is much better described but I now see two problems in these figures. First, I think that during the co-precipitation of dolomite and phyllosilicates (orange zone), the $\Delta^{25}\text{Mg}_{\text{fluid}}$ should increase. This is the case in Figure S6B but not for other calculations. In addition, if this corresponds to a Rayleigh fractionation, I would expect more isotopic variations for Mg loss > 0.9. Finally, I am not sure it is really appropriate to compare the final value $\Delta^{25}\text{Mg}_{\text{fluid}}$ in figure 7 to cation exchange pool isotopic composition. I would expect the value of the exchange pool isotopic composition to represent an average value integrating the last few % of Mg loss.

2- It seems that the figures 4 & 5 do not really indicate the same mixing proportion of carbonates and phyllosilicates, especially for the EDTA and CH_3COOH species.

3- I still find a bit strange to cite papers that are still currently under review as it is difficult to evaluate their connexion with the present work. As a reminder, provide proper citations when such studies will be properly published.

4- I have pointed out mistakes in referring to figures or captions that are not corrected in this new version. Please pay a special attention and correct these issues. It can be difficult to follow if we do not know which figure you are really discussing.

Detailed comments

Results

L156: Fig. 2B is Fig 2C. This was not corrected.

L160: “Figs. 2C and 2DP” is still wrong. Seems to be “Figs. 2C and 2E”.

L162: Reference to fig. 2C is incorrect. Should be 2D.

L163: Reference to fig. 2D is incorrect. Should be 2E.

L239: This is not “grams”. μg .

L240: “Therefore, the influence of lithological heterogeneity on the representativeness of the chemically extracted values is expected to be small.” Not sure to agree for Mg, as about 0.1 ‰ variability is seen in Ryugu Mg isotopic composition (Bizzarro et al., 2023). Such variations are explained by the diverse amount of carbonates within each aliquot, demonstrating the effect of taking only small aliquot. However, this is rather small compared to the total fractionation observed here.

L262: I would recommend keeping only 2 significant digits for consistency.

L264: I agree when comparing with literature data. However, the range for Ryugu in the Bizzarro et al., 2023 study is roughly the same as for Orgueil. Even though most of the Orgueil analyses are on the heavier side, one measurement is almost as light as the lightest of Ryugu (respectively -266 ppm and -286 ppm).

Discussion

L279: Good to give the error. However, I would keep the same number of figures after coma, “-1.34 ± 0.02 ‰”.

L291: Do you mean serpentine instead of serpentinite?

Samples and Methods

L602-603: Please also give the sample cone detail.

Figures

Fig. 4: EDTA is supposed to extract preferentially calcite. You said in the caption of the figure that fractionation coefficient for dolomite and calcite is different. It would be interesting to give numbers as the high $\delta^{25}\text{Mg}$ compared to the mixing line with dolomite could be easily explained by a mixing with calcite that should give a higher $\delta^{25}\text{Mg}$ because of their smaller $\Delta^{25}\text{Mg}_{\text{carb-sol}}$ (L753).

Interestingly, the EDTA is supposed to be very selective on carbonates (at least does not exchange/dissolve with phyllosilicates). However, the Mg-isotopic composition of the EDTA fraction is already quite heavy compared to the Mg-isotopic composition of carbonates. How do you understand this point? Does this mean that the standard phyllosilicates used in table S1 are not properly representing the one of the meteorites?

Fig. 5: Please give the reference of the HCl extraction. Please also remind the reader that “the HF/HClO₄ fractions are not shown on the Ca/Mg plot because Ca was at the lower limit of detection” as written in the text.

Tables

Table 2: Can you add the Mg/(Mg+Fe) for breunnerite, as already request before.

Supplementary materials

Please be consistent between the use of Ma and Myr.

L142: It would be clearer saying $\alpha(^{26}\text{Mg}/^{24}\text{Mg})$ for instance since I did not pay attention that this was not $\alpha(^{25}\text{Mg}/^{24}\text{Mg})$ as given in Fig. 7 or Fig. S6, and it is slightly confusing.

L161: I would not say “slightly older” when the age derived by McCain et al. is more than half the one Yokoyama et al.

L171: “²⁶Al” with 26 as superscript.

L171: The ²⁷Al/²⁴Mg ratio of carbonates, especially from Mg-bearing carbonates would be low, further supporting a low ²⁶Mg-excess if any.

L191: Even if you were able to do high precision Mg isotope measurements, I am not sure that you could really derive some chronological information as most of your extracts seem to be mixing of phyllosilicates and carbonates. Disentangling such mixing might bring large uncertainties.

Assessing partial dissolution of phyllosilicates and carbonates during the sequential leaching experiment: Please be consistent between “A106 and C107” or “A0106 and C0107” in the rest of the text.

L224: I am not sure that the supplementary material will be further edited. So please check that sign and figures are on the same line.

L353: “half the rates for (C) and (D)”. Not very clear. Do you mean removal rate of carbonates if half the one of phyllosilicates?

Supplementary Table S1: EDTA-Na shows almost no reaction with phyllosilicates. However, Figs. 3 and 4 show that Mg-isotope composition is in between phyllosilicates and carbonates. Does this the result of very low Mg content of carbonates or does this mean that phyllosilicates of Ryugu act different from terrestrial ones?

L397: Bizzarro et al., 2023 is now published.

Replies to Reviewer's comments on the manuscript

NCOMMS-23-47707B

We appreciate the constructive comments on our manuscript (# NCOMMS-23-47707B) entitled “**Brunnerite grain and magnesium isotope chemistry within cation-partition dynamics during aqueous alteration of asteroid Ryugu**” (by Yoshimura et al.). We appreciate your constructive comments and have revised the manuscript based on your advices. The changes we have made are indicated in red in the revised manuscript/supplementary information. We believe that the overall context of the paper is now clearer.

General comments

1- Figure 7. The model is much better described but I now see two problems in these figures. First, I think that during the co-precipitation of dolomite and phyllosilicates (orange zone), the $\Delta^{25}\text{Mg}_{\text{fluid}}$ should increase. This is the case in Figure S6B but not for other calculations. In addition, if this corresponds to a Rayleigh fractionation, I would expect more isotopic variations for Mg loss > 0.9 . Finally, I am not sure it is really appropriate to compare the final value $\Delta^{25}\text{Mg}_{\text{fluid}}$ in figure 7 to cation exchange pool isotopic composition. I would expect the value of the exchange pool isotopic composition to represent an average value integrating the last few % of Mg loss.

→ Thank you for your important remarks regarding the isotope fractionation calculations. As you pointed out, there are some missing considerations and I should have corrected the factor of the Rayleigh effect on the Mg residual at the start of the dolomite reaction. The figure and explanation, which I modified on the advice of my colleague, Dr. Chisato Yoshikawa, an isotope modeler, are shown below. Following the previous calculation concept, the precipitation rates (Mg removal rates) of phyllosilicate and dolomite have been adjusted to avoid large changes in the $\Delta^{25}\text{Mg}$ values in dolomite.

The revised calculation ensures that the total dissolved Mg distributed to secondary minerals is 100% for a sum of phyllosilicate, dolomite, and exchangeable Mg. Approximately 5-7.5% was distributed to exchangeable Mg in the previous version of the calculation, but this estimate has been revised. The amounts of the residual dissolved Mg^{2+} (i.e., exchangeable Mg) was revised to 1.2-4.5%, but the amount of exchangeable Mg was calculated relative to the amount of dolomite, so three representative conditions were set up for the calculation in Supplementary Figure S6. See also the revised Supplemental Material. In all cases, the results do not differ significantly from the previous results in that Mg partitioning into the phyllosilicate progresses from the early stages of aqueous alteration and dolomite precipitates in the later stages. In Figure 7, which has been modified in accordance with your advice, the Mg isotopic ratio increases in the dolomite precipitation interval indicated in orange. The Mg precipitation rate was varied so

that this increasing gradient is not too extreme, i.e., the Mg isotopic heterogeneity of the dolomite is not too large. Whether $\delta^{25}\text{Mg}$ homogeneity or heterogeneity of dolomite is practically valid can be tested in the future by directly measuring the Mg isotopic composition of multiple particles. The results of the sensitivity experiments for the model calculations, including the change in precipitation rate, are shown as Supplementary Figure S6.

Figure 7. Model of Mg isotopic changes during aqueous alteration on Ryugu.

Supplementary Figure S6. Models of Mg partitioning calculated from changes in magnesium isotopic ratios.

2- It seems that the figures 4 & 5 do not really indicate the same mixing proportion of carbonates and phyllosilicates, especially for the EDTA and CH_3COOH species.

➔ The EDTA and acetic acid extractions are treatments intended to selectively leach carbonates. However, in response to a previous peer review comment, we had added in the text and caption that there is an effect of partial dissolution of phyllosilicates in these extracts. The significance

of this figure in the discussion was to illustrate the endmember composition of the leaching solutions and to demonstrate the superiority of the individual analysis of carbonate particles, so I added a further note on this subject.

3- I still find a bit strange to cite papers that are still currently under review as it is difficult to evaluate their connexion with the present work. As a reminder, provide proper citations when such studies will be properly published.

→ The citation of Takano et al. was in the final stage of peer review, but it has been officially accepted, so we have revised it.

4- I have pointed out mistakes in referring to figures or captions that are not corrected in this new version. Please pay a special attention and correct these issues. It can be difficult to follow if we do not know which figure you are really discussing.

→ We have double-checked and corrected the citation of the chart.

Detailed comments

Results

L156: Fig. 2B is Fig 2C. This was not corrected.

L160: “Figs. 2C and 2DP” is still wrong. Seems to be “Figs. 2C and 2E”. L162: Reference to fig. 2C is incorrect. Should be 2D.

L163: Reference to fig. 2D is incorrect. Should be 2E.

L239: This is not “grams”. µg.

→ We have made corrections.

L240: “Therefore, the influence of lithological heterogeneity on the representativeness of the chemically extracted values is expected to be small.” Not sure to agree for Mg, as about 0.1 ‰ variability is seen in Ryugu Mg isotopic composition (Bizzarro et al., 2023). Such variations are explained by the diverse amount of carbonates within each aliquot, demonstrating the effect of taking only small aliquot. However, this is rather small compared to the total fractionation observed here.

→ We agree with this point and have added the following text:

“However, there is a variation of ~0.1 ‰ in the Mg isotopic composition of Ryugu (Bizzarro et al., 2023). Such variation is explained by the varying amount of carbonates in each aliquot (Moynier et al., 2022), indicating the effect of taking only small aliquots.”

L262: I would recommend keeping only 2 significant digits for consistency.

→ The values reported by Bizzarro et al. in their high-precision analysis were quoted, but the values have been changed to match the digits in this study.

L264: I agree when comparing with literature data. However, the range for Ryugu in the Bizzarro et al., 2023 study is roughly the same as for Orgueil. Even though most of the Orgueil analyses are on the heavier side, one measurement is almost as light as the lightest of Ryugu (respectively -266 ppm and -286 ppm).

→ Indeed, a similar low value is reported for Orgueil by Bizzarro et al. We have revised the value to clarify the difference from the literature value:

“~ they are slightly lighter than most literature data of Ivuna-type (CI) and other carbonaceous chondrite groups. Most of the Orgueil data reported by Bizzarro et al. (2023) also agree with the literature carbonaceous chondrite values, although one measurement is nearly identical to the lowest value of Ryugu.”

Discussion

L279: Good to give the error. However, I would keep the same number of figures after coma, “ -1.34 ± 0.02 ‰”.

L291: Do you mean serpentine instead of serpentinite?

→ We have made corrections.

Samples and Methods

L602-603: Please also give the sample cone detail.

→ We have added the sample cone type:

“We performed Mg isotope analysis with a nickel sampler cone and a high-sensitivity X-skimmer cone.”

Figures

Fig. 4: EDTA is supposed to extract preferentially calcite. You said in the caption of the figure that fractionation coefficient for dolomite and calcite is different. It would be interesting to give numbers as the high $\delta^{25}\text{Mg}$ compared to the mixing line with dolomite could be easily explained by a mixing with calcite that should give a higher $\delta^{25}\text{Mg}$ because of their smaller $\Delta^{25}\text{Mg}_{\text{carb-sol}}$ (L753).

Interestingly, the EDTA is supposed to be very selective on carbonates (at least does not exchange/dissolve with phyllosilicates). However, the Mg-isotopic composition of the EDTA fraction is already quite heavy compared to the Mg-isotopic composition of carbonates. How do you understand this point? Does this mean that the standard phyllosilicates used in table S1 are not properly representing the one of the meteorites?

→ As for EDTA, it is a reagent used with the intention of selectively extracting calcite or aragonite, so we anticipated to use it for extracting calcite from Ryugu. However, because the reactivity of Ryugu's phyllosilicates was higher than the earth's clay mineral experiments used in Supplementary Table S1, it is thought that even the weakly acidic conditions of EDTA-2Na caused a slight dissolution of the silicate minerals. This was due to the difference in physical

properties from the clay minerals on the earth. As you pointed out in another comment, the Mg content of calcite is much lower than that of dolomite, so even a small amount of phyllosilicate dissolution would have affected $\delta^{25}\text{Mg}$ values. Although the chemical extraction itself was difficult, we have changed the text to emphasize the advantages of the method already shown by Bizzarro et al. (2023) to estimate endmembers by analysis of multiple samples and the direct measurement of Mg isotope ratios of carbonates at the particle level, which we have demonstrated in this study:

“Because of the high reactivity of Ryugu's phyllosilicates, it is difficult to obtain carbonate $\delta^{25}\text{Mg}$ by chemical extraction because even EDTA causes partial dissolution, so direct measurement of the isotope ratio of the microparticles is an effective technique.”

Fig. 5: Please give the reference of the HCl extraction. Please also remind the reader that “the HF/HClO₄ fractions are not shown on the Ca/Mg plot because Ca was at the lower limit of detection” as written in the text.

→ We have added that HCl is from Yoshimura et al. (2023), with the addition that Supplementary Table S5 shows the raw data and that Ca is below the lower limit.

Tables

Table 2: Can you add the Mg/(Mg+Fe) for breunnerite, as already request before.

→ My apologies, I have added the data.

Supplementary materials

Please be consistent between the use of Ma and Myr.

→ They have been united in Myr.

L142: It would be clearer saying $\alpha(^{26}\text{Mg}/^{24}\text{Mg})$ for instance since I did not pay attention that this was not $\alpha(^{25}\text{Mg}/^{24}\text{Mg})$ as given in Fig. 7 or Fig. S6, and it is slightly confusing.

→ The text and figures show only $^{25}\text{Mg}/^{24}\text{Mg}$, but $^{26}\text{Mg}/^{24}\text{Mg}$ is cited only here in this section, so isotopes were added for the individual values.

L161: I would not say “slightly older” when the age derived by McCain et al. is more than half the one Yokoyama et al.

→ The text has been changed.

L171: “²⁶Al” with 26 as superscript.

→ The text has been corrected.

L171: The $^{27}\text{Al}/^{24}\text{Mg}$ ratio of carbonates, especially from Mg-bearing carbonates would be low,

further supporting a low ^{26}Mg -excess if any.

→ We agree with this point, and the description has been added accordingly.

L191: Even if you were able to do high precision Mg isotope measurements, I am not sure that you could really derive some chronological information as most of your extracts seem to be mixing of phyllosilicates and carbonates. Disentangling such mixing might bring large uncertainties.

→ We agree with your comment and have removed the text.

Assessing partial dissolution of phyllosilicates and carbonates during the sequential leaching experiment: Please be consistent between “A106 and C107” or “A0106 and C0107” in the rest of the text.

L224: I am not sure that the supplementary material will be further edited. So please check that sign and figures are on the same line.

→ These has been corrected.

L353: “half the rates for (C) and (D)”. Not very clear. Do you mean removal rate of carbonates if half the one of phyllosilicates?

→ As you pointed out, we intended that the Mg removal rate (precipitation rate) of dolomite relative to that of phyllosilicate. The caption has been revised along with the revised figure.

Supplementary Table S1: EDTA-Na shows almost not reaction with phyllosilicates. However, Figs. 3 and 4 show that Mg-isotope composition is in between phyllosilicates and carbonates. Does this the result of very low Mg content of carbonates or does this mean that phyllosilicates of Ryugu act different from terrestrial ones?

→ As you point out, it is an effect of both: low Mg content in calcite and mixing. The latter, mixing is due to the higher reactivity of Ryugu's phyllosilicates, in particular, to acids than those found on Earth; EDTA-Na was used with the intention of selectively extracting calcite and the subsequent acetic acid with the intention of selectively extracting dolomite. As for calcite, its Mg content is low and therefore susceptible to even slight dissolution of phyllosilicates.

L397: Bizzarro et al., 2023 is now published.

→ We have added the publication information.